# When Politics Gets Personal: Students’ Conversational Strategies as Everyday Identity Work

**DOI:** 10.3390/bs15060835

**Published:** 2025-06-19

**Authors:** Toralf (Tony) Zschau, Hosuk Lee, Jason Miller

**Affiliations:** 1Department of Sociology & Human Services, University of North Georgia, Dahlonega, GA 30597, USA; 2Institute for Environmental & Spatial Analysis (IESA), University of North Georgia, Dahlonega, GA 30597, USA; hosuk.lee@ung.edu; 3Department of Psychological Sciences, University of North Georgia, Dahlonega, GA 30597, USA; jason.miller@ung.edu; 4Department of Psychological Sciences, University of Tennessee, Chattanooga, TN 37403, USA

**Keywords:** identity discomfort, interpersonal communication, student political discourse, identity verification, political polarization

## Abstract

Political polarization in the United States has made conversations across ideological divides increasingly difficult to navigate. This study explores how students at a regional university in the southern U.S. experience and manage these challenges. Based on in-depth interviews with 30 students from diverse social and political backgrounds, we identify four key conversational strategies: disengagement, negotiation, context adaptation, and information processing. Rather than viewing these as surface-level techniques, we argue they represent deeper identity management processes aimed at reducing the social and cognitive risks of political disagreement. Drawing on Self-Categorization Theory and Identity Control Theory, we show how these strategies reflect efforts to maintain identity coherence and manage relational stakes when political identity becomes salient. Our findings suggest that while these strategies may help students avoid conflict in the moment, they may also limit deeper engagement across divides. We conclude by discussing the implications of these findings for dialog practice, highlighting the importance of fostering tolerance for identity discomfort and helping students rediscover the common bonds that unite them across political differences.

## 1. Introduction

Americans are increasingly sorting themselves into opposing political camps, marked not just by ideological disagreement but by a growing unwillingness to engage with those on the other side. According to a Pew Research Center study, only 17% of Americans feel very comfortable discussing politics with strangers, while nearly half (48%) report feeling uncomfortable doing so. Moreover, roughly half of respondents describe such conversations with political opponents as frustrating or stressful—especially among those most politically active ([50]). Over the past decade, this discomfort has translated into a growing avoidance of political conversations across ideological divides ([65]). More recent research suggests that this avoidance is fueled not only by ideological conflict but by the perception that such conversations are emotionally exhausting, socially risky, or unlikely to change anyone’s mind ([26]; [58]). Political conversations, when they do occur, are rarely planned. Instead, they tend to emerge spontaneously—often without clear objectives, structure, or even an explicit desire to persuade ([17]). Despite their informality, these conversations are shaped by powerful cultural norms that dictate what can and cannot be said in “polite” political discourse ([20]). Topics tied to identity—such as race, gender, sexuality, or immigration—are often seen as too sensitive to bring up, even as they remain central to the nation’s most pressing political questions. While avoiding these conversations may help preserve social comfort in the moment, it also risks silencing the very issues that most urgently require open engagement ([37]). Recent research suggests that this pattern of avoidance is not limited to isolated instances but has become a common feature of everyday life—persisting despite growing public and scholarly interest in fostering more constructive engagement across political divides ([58]; [26]).

Scholars have increasingly argued that the deepening political divide in the United States reflects not just ideological and affective polarization, but also a growing epistemic divide over what counts as fact, evidence, and truth. This so-called “post-truth” dynamic has been linked to the spread of disinformation, the rise in conspiratorial thinking, and the erosion of shared factual ground in public discourse ([4]; [5]; [42]). Some scholars have gone so far as to frame this as an epistemological crisis undermining democratic norms of reasoned debate and mutual understanding ([5]; [42]). Other recent work further suggests that misperceptions of ideological opponents—such as overestimating how extreme or hostile the “other side” really is, or underestimating their commitment to democratic values—can amplify these divides, reducing people’s willingness to engage at all ([39]; [18]).[note 1] While these analyses offer important insights into structural and cultural shifts in political communication, our focus in this study is more micro-level: we are less interested in adjudicating claims about “post-truth” dynamics and more concerned with how ordinary people—specifically college students—experience and navigate these divides in their everyday interactions. By foregrounding participants’ own meaning-making processes, we aim to contribute to a more grounded understanding of how polarization is lived, felt, and managed in face-to-face and small-group political conversations.

In this paper, we seek to better understand how young adults navigate the challenges of political conversation in an increasingly divided society. Grounded in a social constructionist perspective, we explore two guiding questions: (RQ1) How do college students describe their experiences of avoiding or engaging in political conversations across ideological divides? and (RQ2) Under what conditions do they feel more open to engaging in these conversations? To address these questions, we drew on in-depth interviews with students at a regional university in the American South, focusing on how they described navigating, managing, or withdrawing from political conversations with both like-minded peers and those holding opposing views. While students’ accounts reflected a range of experiences, their strategies tended to cluster around four recurring patterns: disengagement, negotiation, context adaptation, and information processing. Closer examination of these patterns suggests that these surface-level strategies reflect deeper identity management processes, particularly in moments when political identity is activated by broader social dynamics or specific conversational triggers.

We argue that these strategies can be best understood through the combined lens of Self-Categorization Theory and Identity Control Theory. While Self-Categorization Theory helps explain how political identities may become salient or “primed” in certain contexts, Identity Control Theory offers a framework for understanding the ongoing identity work students perform to maintain a coherent sense of self in the face of social and relational risks. Specifically, these strategies—including avoidance, modification, and reinforcement—appear to function as adaptive responses aimed at managing identity discomfort in politically charged interactions. By mapping and analyzing students’ lived experiences, our findings suggest that conversational strategies are not merely rhetorical tactics, but practical identity management efforts shaped by situational and broader social forces. While necessarily tentative and exploratory, we believe these insights offer a useful starting point for rethinking how people might be supported in engaging more constructively across political divides.

### 1.1. Conversations Across the Aisle

Political conversations are increasingly linked to negative emotions and chronic stress, as individuals report heightened anxiety, frustration, and fatigue when discussing political issues ([25]). A central factor driving this emotional strain is the widening ideological divide between the political left and right in the United States. Political polarization has been steadily intensifying over the past several decades, with recent studies suggesting it has reached unprecedented levels ([48]). Given the emotional toll these conversations can carry, it is unsurprising that many individuals have become more selective about whom they engage with—often preferring to speak only with those who share their beliefs and avoiding interactions with those who do not.

Empirical research confirms that individuals tend to engage in political conversations primarily with those who share their ideological views, reinforcing what are often referred to as “echo chambers.” A large-scale analysis of 3.8 million Twitter users found that people rarely interact with ideologically different others when discussing political topics ([3]). Notably, this avoidance is specific to political content—individuals with opposing political beliefs still interact across lines in non-political contexts. This suggests that ideological disengagement is not rooted in a general aversion to the other side, nor in a lack of social capability, but rather in a conscious decision to avoid political disagreement. While Republicans in the study showed a stronger tendency toward intentional polarization, the pattern was also pronounced among political liberals.

Research also highlights the consequences of intentional polarization. In a classic study, [46] ([46]) found that when individuals discuss controversial issues exclusively with ideologically similar others, their attitudes tend to become more extreme. In contrast, exposure to opposing viewpoints can lead to more moderate positions. This phenomenon—known as group polarization—appears to be driven by repeated exposure to affirming perspectives, which strengthens individuals’ positive evaluations of those attitudes over time ([7]). As people increasingly seek out like-minded political communities, this reinforcement loop deepens ideological divides. When liberals only engage with other liberals, their views are likely to shift further left; when conservatives do the same, their positions move further right. Over time, this dynamic contributes to a widening gap between the two sides of the political spectrum.

Sociological research further supports the idea that engaging only with ideologically similar others can intensify political attitudes. The spiral of silence theory suggests that public opinion tends to coalesce around a perceived consensus, as individuals align their views with those of their social environment ([47]). People are more likely to suppress opinions that diverge from the dominant narrative and more inclined to voice those that reinforce it, leading to a reinforcing cycle of conformity. As dissenting views are silenced and mainstream perspectives are echoed, the perceived consensus grows stronger—and often more extreme. This process is especially evident in online spaces, where ideologically homogenous groups form echo chambers that discourage dissent ([27]). Within these groups, individuals tend to express ideas they believe will be positively received, reinforcing existing norms and amplifying more radical views. Over time, the absence of opposing perspectives allows extreme ideologies to take deeper root, under the illusion that they are universally shared.

While many theories attempt to explain the roots of political polarization, there remains little consensus on exactly why the divide persists or how it might be bridged. Interpretations vary widely across disciplines, and the complexity of political identity makes it difficult to isolate a single cause. In light of this, we adopt a grounded, qualitative approach to explore how individuals navigate the social and emotional terrain of political discourse. Specifically, we investigate why people often avoid conversations across ideological lines—and under what conditions they might feel more open to engaging with those who hold opposing views.

### 1.2. Political Identities

Political identities represent one of the most socially and psychologically charged forms of group affiliation in contemporary public life. Like other social identities—such as race, religion, gender, or nationality—political identities serve as cognitive and emotional frameworks through which people interpret the world, align themselves with social groups, and make sense of their place in society ([35]). These identities typically include partisan affiliations (e.g., Democrat or Republican), ideological orientations (e.g., liberal or conservative), and issue-based commitments (e.g., pro-life, environmentalist, gun rights advocate). Political identities are especially powerful because they often cut across other social categories and are closely linked to people’s moral values, worldview, and sense of belonging ([39]; [28]).

While individuals hold multiple identities simultaneously, political identities tend to rank high in salience—particularly in contexts where political conflict, media framing, or social cues make them more personally relevant ([41]). Research in political psychology suggests that political identities can become “chronically accessible” over time, shaping not just voting behavior but also everyday social judgments, trust in information, and emotional responses to others ([35]; [39]). This salience is reinforced by the way politics is socially and institutionally organized in the United States. Major media outlets, political campaigns, and social networks frequently frame public issues as binary contests between competing partisan groups, encouraging individuals to view themselves and others primarily through a political lens ([36]; [56]). As a result, political identities often take on a “mega-identity” status, absorbing and amplifying other social affiliations and becoming central to how individuals navigate social and political life ([39]).

These identity dynamics shape how people engage in political conversations. As [16] ([16]) demonstrated, individuals tend to respond more positively to members of their political ingroup and more negatively to those in the outgroup—often based on identity cues alone. This ingroup favoritism extends beyond policy preferences to include moral evaluations, with individuals often perceiving their own political group as not only ideologically different but morally superior to the opposition ([6]; [8]; [32]). These dynamics can make cross-ideological dialog feel emotionally charged or socially risky, leading people to approach difficult conversations only with those they perceive as ideologically aligned while avoiding or shutting down interactions with those they view as part of the opposition.

#### 1.2.1. Political Identities via Social Identity Theory

Social Identity Theory (SIT) provides a foundational framework for understanding why political identities become such powerful social markers. According to SIT, individuals derive part of their self-concept from their membership in social groups, including political parties and ideological communities ([60]). These group memberships offer not just a sense of belonging but also a source of self-esteem, motivating people to view their own groups (ingroups) in favorable terms while perceiving rival groups (outgroups) as inferior or threatening ([32]; [8]). In the political realm, this manifests in patterns of ingroup favoritism and outgroup derogation, where individuals elevate the moral, intellectual, or cultural status of their own political side while dismissing or vilifying those on the other side ([6]). Such dynamics help explain why political disagreements often feel personal or emotionally charged, as challenges to one’s political group can be experienced as attacks on the self. SIT highlights that these identity-driven evaluations are not just rational assessments of policy differences, but emotionally laden judgments rooted in the human need to protect and elevate the social groups with which one identifies.

#### 1.2.2. Political Identities via Self-Categorization Theory

While Social Identity Theory (SIT) explains why group attachments form, Self-Categorization Theory (SCT)—which builds on SIT—elaborates on when and how particular identities, such as political ones, become situationally salient ([61]; [62]). SCT posits that people hold multiple social identities, but only some become behaviorally relevant depending on situational cues, perceived group boundaries, and social comparisons. Political identities become especially salient when the social environment highlights partisan distinctions, such as during election cycles, political debates, or in media coverage that frames issues as binary contests between ideological camps ([30]; [53]). The way U.S. media and political institutions frame public discourse—consistently labeling politicians as Democrat or Republican and emphasizing “us versus them” narratives—further reinforces this salience ([36]; [56]). When political identity becomes activated in this way, individuals are more likely to categorize themselves and others along partisan lines, shaping how they interpret information, assess conversational risks, and choose whether or not to engage. SCT thus offers insight into why political conversations can feel immediately polarized or adversarial, even before substantive issues are discussed.

#### 1.2.3. Political Identities via Control Theory

Whereas SIT and SCT focus on group belonging and identity salience, Identity Control Theory (ICT) shifts attention to the ongoing, interactional work individuals perform to maintain identity coherence in social settings ([11]; [14]). ICT conceptualizes identity as a self-regulating system in which individuals seek to align external feedback with their internalized identity meanings. This process, known as identity verification, involves continuously monitoring how one is seen by others and adjusting behavior to preserve alignment between self-perception and social perception ([13]). When this verification process is disrupted—such as when individuals receive feedback that conflicts with their internal identity standards—it can trigger attempts to restore alignment. ICT research has identified a range of corrective responses, including behavioral adjustment, increased self-assertion, withdrawal from interaction, or cognitive reappraisal of the situation ([14]). While ICT has been widely applied in studies of role-based and interpersonal identities, its application to political identity dynamics—especially in the context of political disagreement—remains underexplored.

#### 1.2.4. Summary: Why Everyday Political Identity Matters

While scholars across disciplines have proposed a range of explanations for the persistence of political division, there remains little consensus on how these divides actually unfold in the flow of everyday social interaction. Much of the existing research has focused on large-scale systems—such as media ecosystems, partisan institutions, or psychological predispositions—leaving fewer studies that examine how political identities shape ordinary encounters across ideological lines. Yet it is often in these everyday moments—whether around dinner tables, in classrooms, or in casual conversations—where political identities are enacted, challenged, or managed in personally meaningful ways.

Understanding the nature of this identity work is not just an academic exercise—it is a crucial step toward recognizing the small, often overlooked ways people try to hold themselves—and their relationships—together in divided times. Whether this deeper understanding can help us build better tools for navigating these divides remains an open question. But before we turn to that possibility, we first take a closer look at the identity processes that appear to shape these everyday conversational strategies—because it is in that deeper unpacking that the true stakes of this work begin to come into view.

## 2. Materials and Methods

### 2.1. Participants

The study involved 30 participants recruited from a regional university in the South, with a balanced representation across key demographic categories (see Table 1 for a summary). The sample was evenly split by gender (15 male, 15 female) and political orientation (10 Conservative, 10 Liberal, and 10 Other), ensuring the creation of three gender-balanced ideological groups. Racial diversity was also ensured, with 8 African American, 10 Caucasian, 6 Hispanic/Latino, and 6 Multiracial participants. Participants’ ages ranged from 18 to 24, with an average age of 21. This diverse composition was designed to enable a thorough investigation into how various demographic factors may influence conversations across ideological divides.

### 2.2. Sampling and Recruitment Strategy

*Sampling:* A purposive sampling strategy was employed to explore what makes conversations across the ideological divide easier or more difficult, focusing on two key criteria: gender and political orientation. Participants were selected to ensure an even distribution of males and females, as well as a balanced representation across political ideologies. Political orientation was determined based on participants’ open-ended responses in a screener survey, which then were coded into one of three categories: Conservative, Liberal, or Other. These categories were selected as they best captured the range of political perspectives within the sample, allowing for meaningful comparisons across the political spectrum.

*Recruitment:* Participants were recruited through multiple channels, including a university-wide research system, word of mouth, and classroom outreach. Before the interviews, participants completed a screener survey that collected demographic information and assessed their current level of engagement in political conversations. Based on this screener, participants were categorized by gender and political orientation to ensure a balanced and diverse sample. Informed consent was obtained from all participants, and pseudonyms were assigned to protect their identities. As an incentive, students were compensated for their participation.

### 2.3. Interviews and Data Collection

A semi-structured interview approach with unstructured elements was used to allow flexibility in exploring participants’ experiences. The four main topics guiding the interviews were *nature of conversation*, *context of conversation*, *outcomes of conversation*, *and network dynamics*. Our interview guide began with broad, open-ended questions that allowed participants to describe conversations they had with “the other side” in their own words and describe any experiences that came to mind. When participants initially described only general or non-political interactions, we used follow-up prompts to encourage them to reflect more specifically on politically charged conversations or moments of political disagreement. This progressive narrowing is consistent with best practices in qualitative interviewing, which balance participant-led meaning-making with interviewer guidance through follow-up questions, probes, and clarifications designed to focus the conversation on key areas of interest ([52]; [9]). Examples of “starter” questions included:“When was the last time you had a conversation with someone who does not share your political views? What was that conversation like?” (nature of conversation)“Where do you usually have these conversations? Are there any places where you find it easier to discuss political topics?” (context of conversation)“How did these conversations affect your views? Were there any lingering emotions? How has this experience shaped your willingness to engage in future political conversations?” (outcomes of conversation)“How do your friends and family talk about people with opposing political views? How does your ____ (name of network) perspective influence your willingness to engage with others?” (network dynamics)

This approach was designed to allow for structured comparisons across gender and political orientation while giving participants the freedom to elaborate on their personal experiences. By balancing structure with openness, the study captured both commonalities and nuances in how participants engage in political conversations. Interviews were conducted during the fall semesters of 2020 and 2021, averaging 55.62 min each and totaling approximately 31 h and 35 min across the sample. The flexible format enabled interviewers to adjust questions based on participants’ responses. All interviews were recorded and transcribed using QSR’s AI-based transcription service (QSR International, Melbourne, VIC, Australia), followed by hand-editing for accuracy. The final transcripts were then imported into NVivo 12.0 for data analysis.

### 2.4. Data Analysis

*Initial Coding:* Initial coding began inductively, with early rounds of open and axial coding conducted on subsets of the data (n = 8, 10, and 16 interviews, respectively). These rounds produced a broader set of candidate themes, including conflict management strategies, network dynamics, and emotional coping responses. As coding progressed, the team observed considerable thematic convergence, ultimately collapsing these preliminary categories into five higher-order strategies: disengagement, negotiation, context adaptation, information processing, and emotional management. NVivo 12.0 (QSR International, Melbourne, VIC, Australia) was used in this phase to organize transcripts, generate preliminary coding clusters, and assist in the formulation of conceptual definitions ([15]). After de facto thematic saturation, all 30 interviews were re-coded deductively using this five-theme framework. One team member conducted the coding, while a second team member spot-checked and participated in collaborative discussions to ensure consistent interpretation and application of the coding framework. While no formal intercoder reliability metrics were established, this collaborative process helped align coding practices. All final coding was conducted manually, with NVivo serving as a project management tool rather than an automated coding engine.

*Conceptual Refinement and Thematic Coverage:* As the analysis progressed, the emotional management theme—though present in earlier interviews—appeared less consistently across the full dataset and was often woven into other cognitive strategies. For this reason, it was ultimately integrated into the interpretation of the four primary themes rather than retained as a standalone category. These four final themes were prevalent across the majority of participants, with disengagement strategies used by 77%, negotiation by 80%, context adaptation by 70%, and information processing by all participants (100%). Many students used multiple sub-strategies within each theme, reflecting the layered and dynamic nature of their conversational management. These coverage rates reflect the presence of each strategy across the sample, providing a sense of thematic breadth. Importantly, our analysis was guided by the study’s two central research questions, which provided an interpretive frame for both coding and thematic synthesis. Rather than aiming to “test” propositions, our goal was to illuminate how students made meaning of their experiences—and the conditions under which political engagement became possible or foreclosed. However, consistent with broader qualitative research principles, our focus remains on the interpretive richness and variation in how these strategies were described, rather than on quantifying prevalence as in statistical analysis ([54]; [40]). As such, the four main themes and ten subthemes should be understood as representing the most prominent patterns of meaning-making that emerged from participants’ accounts—patterns that help illuminate the complex ways students navigate political conversations across ideological divides, without claiming to offer an exhaustive account of every possible strategy or experience.

*Comparative Analysis:* In addition to coding the transcripts, several comparative queries were run using NVivo to examine whether gender or political orientation influenced how the identified themes manifested. These queries and qualitative analyses aimed to identify potential variations in conversational strategies based on demographic factors. However, the analysis revealed no significant differences across gender or political orientation, suggesting that the identified themes may be shared across a wide variety of groups, regardless of demographic differences. As a result, no specific findings from the comparative analysis will be presented, and the focus remains exclusively on the emergent themes and subthemes from the coding process.

## 3. Results

The findings from this study reveal four main themes that can help explain how students navigate political conversations with those who do not share their political convictions: *Disengagement Processes*, *Negotiation Processes*, *Context Adaptation*, and *Information Processing*. Each theme, along with its respective subthemes, provides insight into the core strategies students employ to manage these interactions. Although these strategies were grounded in participants’ own narratives, their identification and interpretation were shaped by the study’s guiding research questions. These questions provided the analytic frame through which we examined how students described their experiences (RQ1) and the conditions that shaped their openness to political engagement (RQ2). While the four primary themes form the foundation of this analysis, the data also revealed emotional dynamics that warrant brief mention here. In earlier rounds of coding, emotional responses such as stress, frustration, and fear were identified as a distinct theme. However, because these emotional elements were sometimes explicit but often only implied or woven “between the lines” of students’ accounts, this theme was eventually integrated into the broader cognitive strategies presented here.[note 2] Nonetheless, the emotional undercurrent remains evident in several ways. For example, some students described actively seeking emotional support from trusted family members or close friends as a way to process the stress of political conversations. Others reported withdrawing emotionally through distraction strategies such as scrolling on their phones or playing video games to avoid further engagement. Additionally, fear of conflict or even violence emerged in some accounts, with students expressing concerns about physical safety or relational damage when entering politically charged discussions. These observations suggest that, while this analysis emphasizes the cognitive and strategic dimensions of how students navigate disagreement, these processes are often intertwined with emotional concerns, highlighting the complex interplay of cognitive and emotional factors in students’ everyday experiences. Even so, because these emotional dynamics were not consistently foregrounded across all interviews, the focus of this results section will center on the four main themes and their respective subthemes, as they best capture the primary strategies students use to navigate contentious political conversations and differences in convictions (see Figure 1).

### 3.1. Disengagement Processes

Disengagement processes refer to the strategies students use to avoid or withdraw from political conversations they perceive as socially risky, unlikely to yield productive outcomes, or at times cognitively or emotionally taxing. These mechanisms often emerge early in conversations—or even preemptively—as a way to minimize conflict and prioritize self-preservation. Three primary subthemes define these strategies: the “time and energy” excuse, the keeping-the-peace rationalization, and conversation enders. The time and energy excuse highlights how students disengage by citing emotional or physical exhaustion, particularly when conversations feel futile or repetitive. The keeping-the-peace rationalization reflects a deliberate effort to maintain social harmony, especially in close relationships, by avoiding discussions that could provoke conflict or discomfort. Finally, conversation enders involve both verbal and non-verbal strategies—such as brief comments, facial expressions, or changes in tone—that signal disengagement and often bring conversations to a natural or abrupt conclusion. Together, these disengagement strategies enable students to navigate politically charged interactions by sidestepping potential emotional or social consequences without directly engaging in confrontation.

#### 3.1.1. Time and Energy “Excuse”

The time and energy “excuse” is one of the most common disengagement strategies employed by students, though it manifests in different and often nuanced ways. For some, like Melissa, it’s simply “easier to just avoid the topic altogether,” while Mark explains that he “just leaves it alone” when the conversation feels unnecessary. Similarly, Michael shared that he “doesn’t talk to people on the opposite side unless [he has] to,” reflecting a more pragmatic and context-dependent form of disengagement. These examples highlight a passive and reactive approach, where withdrawal is often shaped by situational factors like perceived irrelevance or discomfort. At the same time, some students frame the time and energy “excuse” as a way to prioritize their well-being and avoid what they see as unproductive or repetitive conversations.

Many students use this strategy to avoid conversations they perceive as futile or unlikely to yield meaningful dialog. This approach is often driven by the belief that the other party is unwilling to listen or change their perspective, leading students to prioritize their emotional energy and avoid confrontations they see as pointless. Clarissa shared that she disengages when she feels a conversation might “just turn into… a yelling match,” highlighting a reluctance to escalate conflicts. Similarly, Ashley pointed out that if others “don’t want to learn… then what’s the point?” framing her withdrawal as a response to perceived closed-mindedness. Ella echoed this sentiment, noting how “no matter what you say, they’re just going to argue,” further reinforcing the futility of such exchanges. Samantha added that past attempts to explain her views had been ignored, concluding that “it’s just not worth the time.” Michael summed up this cost–benefit dynamic, sharing that he avoids conversations when they leave him feeling “mentally drained.” Melissa also reflected on the emotional toll, stating that she feels “more comfortable… walk[ing] away from those conversations if [she] need[s] to.” These reflections illustrate how the perception of futility drives disengagement, as students weigh the emotional costs against potential benefits. Rex captures the essence of this strategy best when he states:

“I’m not going to waste my time arguing why trans people deserve rights when someone is clearly not going to change their mind.”—Rex (Male, 18)

Students also use the time and energy “excuse” to prevent further frustration or emotional burnout from repeated attempts to engage in political conversations. This disengagement reflects a deeper sense of exhaustion from trying—and failing—to have meaningful dialogs with individuals who appear unwilling to listen or consider alternative viewpoints. Their decision to step back is often shaped by a history of emotionally draining exchanges. Melissa described how past conversations left her feeling drained, explaining that repeated efforts led to “the same thing over and over again.” Rex echoed this frustration, sharing that he “got tired of it after a while” because discussions always circled back to “the same arguments.” Similarly, Michael noted the toll of feeling unheard, saying, “you just get tired of having to say the same thing over and over.” Mark highlighted how “exhausting [it is] to constantly have to defend your position” against people who refuse to listen. Hazel encapsulated this shared sentiment best, by saying:

“Well, in the past when I’ve tried, it just wasn’t worth the time and energy, and I don’t want to. I don’t even want to try to bring it up because they never seem to understand it from my point of view ever.”—Hazel (Female, 20)

The time and energy “excuse” reflects students’ efforts to step back from conversations they perceive as draining, repetitive, or unlikely to be productive. While students occasionally describe these moments in emotional terms—such as feeling worn out or frustrated—they more often frame them as practical choices to manage their time, energy, and social relationships. Rather than investing effort in conversations they believe are unlikely to change minds or lead to meaningful understanding, students weigh the costs and benefits of engagement and often choose to disengage. This practical orientation toward minimizing conflict or preserving social comfort becomes even more apparent in the next disengagement strategy: keeping the peace.

#### 3.1.2. Keeping-the-Peace

For many students, disengaging from political conversations is not just about avoiding conflict; it is fundamentally about preserving their relationships with friends and family. These conversations are often seen as potentially divisive, with the risk of creating long-term tensions or damaging bonds. Michael explained this pragmatically, noting, “It’s not necessarily so much as I’m intentionally trying to avoid it, but I just know it’s not going to go anywhere.” Similarly, Patrice shared that she often “tr[ies] to stray away from them [political conversations] with family,” illustrating the deliberate effort to prevent friction in close relationships. Nick added that disagreements don’t have to escalate, saying, “At the end of the day, like, we all know that it’s okay to disagree, and you don’t have to fight about it.” This overarching desire to maintain harmony manifests in two main strategies: avoiding conflict to prevent discomfort or hurt and remaining silent or neutral to de-escalate tense situations.

One common approach involves disengaging from conversations to avoid making others feel uncomfortable or hurt. For many students, this is less about suppressing their own views and more about safeguarding their relationships. Taylor reflected on this, explaining that “for the sake of [others’ emotions]” she chooses not to say anything that might upset them. Patrice shared a similar sentiment, emphasizing her efforts not to “hurt their feelings or [make them think] I don’t value their opinion.” Hannah highlighted the emotional stakes, noting that she tries to avoid situations where “tensions [might] rise or some people [might] feel uncomfortable.” Similarly, Hazel described the delicate nature of such conversations, remarking that “we’re all kind of on edge already.” Alyssa added that even passive listening can help avoid conflict, observing that “sometimes when I just try to hear them out, it avoids the conflict.” These examples show how students prioritize maintaining emotional harmony, even if it means sacrificing open dialogue. As Hannah further stressed, the cost of engagement can be high, explaining that she avoids discussions to prevent others from “secretly hating [her]” or starting “any drama.” John, however, probably captures this relationship-saving approach best when he talks about the potential risks in especially uncertain situations:

“It’s… one of the things right now [that] if I open that can of worms, I don’t know the outcome. I don’t know the possible relationships [at risk] … I [try] … not to express my opinions in those situations, like do I keep this environment neutral or do I kind of like sour it by disagreeing or … openly going against others’ beliefs.”—John (Male, 22)

Another way students navigate the complexities of keeping the peace is by remaining silent or neutral in politically charged conversations—especially in discussions with those they care about. By withholding their own perspectives, they seek to diffuse tension and avoid conflict. Brittany described this approach succinctly, stating, “I just kind of let it go. I’m not really going to argue.” Similarly, Lucas shared, “I’d rather just let them believe I’m neutral and stay out of it.” Some students adopt a listening role, choosing to hear others out rather than risk confrontation. Alyssa –stresses that “I kind of just, like, remain silent on those topics. Sometimes when I just try to hear them out, it avoids the conflict.” She adds that with her mother that it is “kind of hard to sort of have [her] opinion” so she “just to keep the peace here.” Clarissa also noted a measured approach, sharing, “We can definitely talk about politics, but we don’t make it a thing where we have to come to a solution.” Rex again captured the essence of this strategy best by reflecting on why he chooses to stay quiet about his views on very contentious political topics:

“I don’t want to say it should happen, but like it should have happened [the storming of the US Capitol on 6 January 2021] … And my reasoning behind that is like our country was founded on kind of rebellious activities like that… So yeah, I tend to kind of keep that to myself.”—Rex (Male, 18)

The keeping-the-peace rationalization goes beyond avoiding discomfort, reflecting a deliberate effort to maintain harmony and protect relationships. By avoiding conflict, remaining neutral, or listening without engaging, students prioritize relational stability over political discourse, carefully weighing the risks of potential tension. This highlights the social and emotional stakes of politically charged interactions. Not all disengagement strategies, however, involve active decisions or spoken rationalizations. In Conversation Enders, students use non-verbal cues—like gestures, tone, or body language—to signal disinterest or discomfort, often ending conversations without direct confrontation.

#### 3.1.3. Conversation Enders

For many students, disengaging from political conversations does not always require direct statements or lengthy justifications. Subtle cues—both verbal and non-verbal—are often enough to signal disinterest or discomfort, enabling disengagement without escalating tensions. These strategies are particularly common when students perceive further discussion as unproductive or contentious. Stephen shared how he disengages when conversations reach an impasse, noting he becomes “done with the argument” once it feels futile. Similarly, John explained how he recognizes when a discussion has lost its value, observing that “they’re not listening to me anymore.” These subtle signals allow students to manage the emotional stakes of political discourse, prioritizing their well-being over prolonged confrontation. Broadly, these conversation-ending strategies fall into two categories: non-verbal cues and verbal strategies.

Non-verbal cues are one of the most common ways students disengage. By relying on gestures, facial expressions, or other body language, they communicate their desire to end the discussion without explicitly saying so. Grace shared an example of this, noting,

“He’ll just kind of give[s] me this look, and I know it’s time to drop it. It’s like we’ve hit a wall, and I’m not getting anywhere.”—Grace (Female, 23)

Similarly, Andrew explained that he often “shrug[s] [his] shoulders or… roll[s] [his] eyes” to show disinterest and avoid further engagement. Some participants described how they pick up on non-verbal cues from others to determine when to disengage. Rex shared that “If someone, like, starts waving their hands all over the place and being really energetic.” He knows this is the end. Samuel echoed this sentiment, explaining that when someone starts “making faces… it’s like, you’re not even listening. You’re just trying to argue,” which leads him to withdraw from the conversation. Others described non-verbal disengagement as a more passive process. Michael noted that at home, political discussions often end without anyone explicitly acknowledging it: “It just turns into one parent making an offhand comment… and then it just ends there.” Similarly, Samantha described a mutual disengagement with her friend after a heated discussion: “We were both kind of like quieter after that and just like were back on our phones.” These examples illustrate how non-verbal cues allow students to manage their emotional energy and disengage from conversations in a way that minimizes direct conflict.

While non-verbal cues are often subtle, verbal strategies can provide a more direct way for students to signal disengagement. These include short, polite phrases or remarks that close the conversation or redirect attention. Patrice described a situation where she disengaged by saying, “Oh, I don’t know what I believe. And that’s OK. Have a good day, sir.” Others—like Taylor—use more subtle verbal hints to signal the desire to end the conversation. He says he likes to

“…slip in a comment here and there, but… keep [his] mouth shut and let them go”.—Taylor (Male, 22)

Some students are even more direct and use explicit verbal strategies to navigate conversations they perceive as unproductive. Mark explained, “Eventually I’m like, OK, he’s too far gone and there’s no way he’s going to believe me,” illustrating his decision to disengage once he determines that further discussion is futile. Lucas described a similar experience during a debate on immigration, noting that he tried to share his perspective but eventually realized his peer “wasn’t open to [his] viewpoint,” leading him to shut down. These dynamics are often bidirectional. Ella described a situation where her mother abruptly ended a conversation by saying she “didn’t want to talk about this anymore,” and Ella simply accepted it, replying, “OK.” These verbal strategies allow students to end conversations gracefully, maintaining a sense of politeness while avoiding deeper conflict. Yet others use humor or casual remarks to redirect conversations. Levi, for example, put it best when he discusses how he subtly shifts the topic to avoid further engagement:

“I probably just kind of, ‘Oh, okay. Yeah, sure, I understand … [and then I say] ‘So this hot dog tastes nasty’ and just go from there.”—Levi (Male, 20)

The conversation-enders strategy highlights how students disengage from political discussions while minimizing tension and preserving relationships. Whether through non-verbal cues like facial expressions, gestures, tone, or verbal strategies such as polite remarks or redirections, these approaches allow participants to navigate the emotional stakes of contentious conversations. Unlike the time and energy excuse, which focuses on preemptively avoiding engagement, or the keeping-the-peace rationalization, which emphasizes relational harmony, conversation enders are often reactive, arising in the midst of dialog as a way to gracefully exit unproductive or emotionally charged situations. Together, these strategies reflect a broader pattern of conversational disengagement, allowing students to manage perceived social risks and navigate politically charged interactions with minimal confrontation or emotional investment.

### 3.2. Negotiation Processes

In contrast to disengagement, where students avoid political conversations entirely, negotiation processes involve strategies used once a conversation is already underway. Rather than stepping back or shutting down, participants described working to manage the flow and tone of these conversations in ways that kept them constructive and socially manageable. These strategies were less about persuading others or resolving disagreement and more about keeping conversations productive, respectful, and contained—preventing them from becoming unmanageable or relationally damaging. Central to these processes was the recognition that political conversations can be difficult but navigable, provided that participants made deliberate efforts to balance expression with social awareness. Students described using strategies such as active listening, agreeing to disagree, and finding common ground to maintain civility, manage conversational tension, and avoid unnecessary escalation. These approaches allowed them to participate without feeling like they had to fully resolve the disagreement or win the argument, while also managing the risk of conversations becoming emotionally charged or socially uncomfortable. Unlike disengagement, which seeks to avoid these conversations altogether, negotiation reflects a practical willingness to stay engaged while minimizing potential social costs—a balancing act that many students described as key to navigating politically charged discussions in their everyday lives.

#### 3.2.1. Active Listening

Active listening emerged as a key strategy for students employed to navigate political conversations with individuals holding differing viewpoints—though it encompassed a range of approaches. For some, it served pragmatic purposes, such as managing tension or avoiding conflict. Alyssa described wanting to understand others’ “actual understanding and standpoint from… and like see how they think of it, like maybe there’s something I’m missing,” though not all found these efforts fruitful. Evan observed that their conversations “get nowhere when someone on either side keeps saying things and they’re just reciting things that they hear on the news,” highlighting the limits of engagement when dialog lacks originality. Tony reflected on the futility of debates that devolve into “name calling” without providing “any insight into the views,” illustrating how such exchanges can feel unproductive. Hannah reframed active listening as a tool for “education” rather than validation, emphasizing its potential to create space for understanding even when agreement is not possible. While students employ a range of different ways to “listen,” two strategies reflect more traditional active listening practices: openness to diverse perspectives and prioritizing respectful dialog.

Many students approached active listening as an opportunity to broaden their understanding by considering viewpoints different from their own. This subtheme emphasizes how they consciously sought to engage with others’ perspectives as a way of fostering empathy and self-reflection. Clarissa shared her openness, stating, “I’m always open to listening to what [her friend] has to say.” Similarly, Hannah remarked, “It’s about learning why they think that way, not just debating them,” demonstrating an approach centered on inquiry and “true understanding” rather than confrontation. Such reflections highlight an intentional effort to listen without judgment, valuing the opportunity to understand others’ beliefs. For some, this openness involved treating conversations as learning experiences that could prompt personal growth. Emily explained that listening to others’ viewpoints sometimes “challenges my own beliefs in a good way,” framing these conversations as opportunities for self-reflection and intellectual engagement. However, Michael probably encapsulated the essence of this strategy best when he stressed the need for empathy and contextual understanding in these types of conversations:

“I just I really try to keep reminding myself as I’m listening to think of it… through their perspective, but also to understand that they were raised that completely different way.”—Michael (Male, 20)

The desire for respect and civility emerged as another key strategy of active listening. Partially reflective of conversational culture in this part of the country, students emphasized the importance of creating an environment conducive to meaningful exchanges. Evan articulated this need, observing that “their conversations get nowhere when someone on either side keeps saying things and they’re just reciting things that they hear on the news,” highlighting the frustration of stagnant dialog. Alyssa shared that she finds it helpful to “stay calm and make sure they feel heard” to improve the quality of discussions, underscoring the role of conversational self-management—and at times emotional regulation—in keeping discussions on track. For others, respectful listening was about validating the other person’s experience while maintaining their own stance. Grace explained, “I try to phrase my points so they feel valued even if we’re arguing,” emphasizing the importance of mutual respect even during disagreement. Emily reflected, “Respectful listening makes it easier to share your perspective later,” suggesting that civility paves the way for more productive dialog. Clarissa summarized this dynamic well, noting, “Listening respectfully doesn’t mean you agree; it just shows empathy,” reframing respect as a foundation for meaningful exchanges. However, Evan—probably better than others—stresses the transformative potential of respectful dialog and active listening when he states:

“But if you can sit down and have a truly educated conversation with someone on the other side of why they believe, what they believe … [and] how they think that’ll help the country or help the situation in the world. And that’s a productive conversation.”—Evan (Male, 24)

Active listening is a multifaceted strategy that students use to navigate political conversations. By embracing openness to diverse perspectives, some engage thoughtfully to broaden their understanding and challenge their beliefs, while prioritizing respectful dialog emphasizes conversational discipline and mutual recognition as essential for meaningful exchanges. However, many students recognize that active listening—while desirable—often fails to yield productive outcomes. As a result, they often turn to a second negotiation strategy: agreeing to disagree. Rather than striving to bridge differences, participants accept them, opting to coexist peacefully with opposing views or disengage from deeper engagement altogether.

#### 3.2.2. Agree-to-Disagree

Agreeing to disagree emerged as a key strategy students used when they recognized consensus or productive discourse was unlikely. This approach allowed them to maintain civility and avoid conflict by accepting differing viewpoints, without the need to resolve them. For some, like Melissa, it was about recognizing when continuing the conversation would be unproductive: “We weren’t really, like, fully listening. We were both just kind of waiting to say our next point and then agreed to just stop.” Similarly, Nick reflected, “We ended on the conclusion that we’ll just like, yeah, we’ll agree to disagree,” showing how this strategy helps bring discussions to a peaceful close. Hazel shared, “I was like, what do you think? And… she gave her answer, and I was like, okay, that’s fine, let’s move on,” emphasizing how agreeing to disagree serves as a way to disengage without further escalation. Or as Samuel put it, “Let’s agree to disagree because it’s better than, you know, trying to push something on someone.” At a substantive level, however, most agree-to-disagree strategies revolve around either a mutual recognition of differing opinions or a more conscious conflict avoidance and emotional regulation.

For many students, agreeing to disagree was a way to acknowledge and respect differing viewpoints while preserving relationships. This subtheme highlights how participants embraced differences without striving for agreement. Ashley described her perspective, saying, “But I came to the realization that, yes, I can hold my opinions, but they can also hold theirs, and that’s okay.” This reflects how she navigates disagreements by recognizing the validity of opposing views, allowing both sides to coexist without conflict. Similarly, Clarissa reflected on how agreeing to disagree helped her maintain friendships, noting, “We can just, you know, reinforce in that each opinion is valid, and that’s how we stay friends.” For her, the strategy strengthens relationships by acknowledging and respecting others’ perspectives rather than forcing agreement. For some, respecting differing opinions served a dual purpose: maintaining connections and ensuring civil conversations. As John put it, “I guess agreeing to disagree is probably one of the easier ways to keep things civil without actually agreeing on anything.” But in the end, for many, agreeing to disagree provided a dignified way to walk away from a conversation that had reached an impasse. Melissa captured this best:

“I’m just going to keep talking when I feel like I’m talking to you a brick wall now, you probably feel the exact same way right now …. We weren’t really, like, fully listening. We were just like continuing to reaffirm our position … obviously we weren’t yelling, but it kind of felt like… [so we decided to] ‘agree to disagree’ and … pat each other on the back and… walk out…”—Melissa (Female, 19)

For many students, agreeing-to-disagree also served as a tool for managing conflict and preventing conversations from escalating emotionally. This subtheme illustrates how students tactfully disengaged when discussions became tense or unproductive. Ella, for example, described how she used the strategy to de-escalate with her mother, saying, “My mom is like I don’t wanna talk about this anymore, and I just agree and leave it at that,” showing how agreeing to disagree helps prevent unnecessary friction. Similarly, Hazel described how she avoided conflict by moving past a disagreement, noting, “I was like, what do you think? And… she gave her answer, and I was like, okay, that’s fine, let’s move on,” demonstrating how the strategy can quickly defuse tension. Nick reflected a similar experience, explaining that, after trying to convey his perspective without success, he and his conversation partner simply “agreed to disagree.” He stressed that he “don’t want to get it” because his friend was “a very hard man to convince.” Whatever the particulars, in the end, this approach offered a dignified exit from conversations that had reached an impasse, allowing students to avoid further escalation. Samuel probably summed it up best when he emphasized that agreeing to disagree is fundamentally a recognition of the limits of engagement:

“Yeah, I usually will say, like, let’s agree to disagree because it’s better than, you know, trying to push something on someone.”—Samual (Female, 18)

Unlike active listening, which focuses on understanding and engaging with different viewpoints, agreeing to disagree involves a range of communicative strategies students use to navigate exit points, allowing them to preserve relationships without escalating conflict. However, negotiation strategies don’t end there. Many students also try to take proactive steps to bridge divides and foster more productive dialog by actively seeking areas of agreement. Rather than simply disengaging, these students focus on finding shared values or principles that can de-escalate tensions and create a sense of connection, even when opinions diverge.

#### 3.2.3. Finding Common Ground

Finding common ground emerged as another key strategy students employed to navigate political conversations. Unlike agreeing to disagree, which often serves as a respectful exit from unproductive discussions, finding common ground emphasizes connection and collaboration. As Hannah’s experience demonstrates, however, this approach can be difficult. This is especially true in cases when the other party does not fully “understand why [the] issue is a big issue” for someone, or worse, is unwilling to try. Despite these challenges, students engaged in various strategies to prevent conversations from becoming hostile or adversarial. For many students, this strategy involved actively seeking areas of agreement, shared values, or common experiences to foster mutual understanding and de-escalate tensions. Mark reflected on how conversations in his political club rarely changed minds but often left participants acknowledging that an opposing viewpoint “made sense.” Similarly, Michael described how disagreements with others eased once they understood “where the motivation came from,” noting that conversations often “just stopped there.” These examples illustrate how students used shared values and a deeper understanding of others’ perspectives to navigate differences. While finding common ground always involved a focus on these connections, students negotiated this in two main ways: using shared experiences as a bridge and focusing on practical compromises to build consensus.

Many students used shared personal experiences or values as a foundation for finding common ground in political conversations. This approach highlights how participants connected across ideological divides by relating their own lives to others. John, for example, described engaging with someone whose views were vastly different from his own: “I had a conversation with this dude who was a conservative… we kind of worked it out… brought our opinions closer.” By finding relatable experiences, he was able to soften ideological differences and create mutual understanding. Similarly, Samuel reflected on how differing life circumstances helped him appreciate another’s perspective: “It necessarily wasn’t… the case of like direct arguing… it was mainly due to our or our circumstances of how we go about choosing our political candidates. Like… my family and I, we live on a farm… her situation is different because she’s a, you know, young woman… dealing with a lot of social oppression… [that’s] why she wants to support Biden.” By contextualizing political decisions within personal experiences, Samuel engaged respectfully despite divergent beliefs. Others found that by starting with a common ground, they could preempt tensions. Ella explained, “It’s more calm if we start by talking about something that we agree on,” emphasizing how agreement can steer conversations away from conflict. Stephen noted, “Well, he’s a really understanding smart guy, and he knows that … on both sides, there is… good points,” underscoring mutual respect and the search for commonalities among his close relationships. In the end, however, the goal of these strategies, as Lucas points out, is about how personal experiences can be used to build bridges and ensure conversations end on the same page:

“People are really divided…when it comes to conversation about [politics]… [conversations] with one of my other friends … sometimes … can be a little rough. But in the end, there’s always gonna be a mutual understanding about where we stand.”—Lucas (Male, 20)

While shared experiences provided a relational foundation, practical compromises allowed students to navigate ideological divides by focusing on actionable solutions or points of agreement. Levi explained how this worked in friendships, sharing that “you don’t want to end the friendship,” so it’s important to “find common ground to talk about again.” In this way, Levi highlighted how preserving relationships often required deliberate compromises. Similarly, Nick described balancing agreement with self-expression during contentious conversations, noting that he would “agree with them” on topics like global warming but still “give [his] two cents.” By aligning with shared beliefs while subtly introducing his perspective, Nick maintained dialog without confrontation. For others, practical compromises meant actively avoiding contentious points and focusing on areas of alignment. Samantha explained, “If one of us disagreed, it was like, OK, that’s fine,” and they shifted to “talk more about what we agreed on.” This approach allowed her to keep conversations positive and productive. Melissa offered a particularly insightful account of how she and others actively negotiated their “common ground.” She described the process as follows:

“… whenever that happens… I go upstairs… do some breathing exercises… Then I go back downstairs. They [my parents] usually do the same thing… [then] we try to come to some sort of… ‘here’s where we agree’… and we kind of leave it at that.”—Melissa (Female, 20)

She also stressed that they both—her parents and her—had to learn “that screaming matches don’t help” and emphasized that sometimes you simply need to “take a step back” before you can “move forward.”

By focusing on common ground, participants were often able to steer discussions toward shared perspectives and manageable takeaways, allowing the conversation to remain constructive even without full resolution. Together with two other strategies—active listening and agreeing to disagree—these negotiation processes enabled conversation partners to create and maintain a more collaborative dialog, at least until an impasse was reached. However, even the most collaborative strategies required flexibility in how they were applied. Students demonstrated this adaptability by adjusting their communication strategies to suit the specific dynamics of each interaction, tailoring their approaches to the context and their particular conversation partners.

### 3.3. Context Adaptation

Context adaptation emerged as a key strategy for students navigating politically charged conversations, requiring them to adjust their communication approaches based on both the setting of the conversation and the characteristics of the people involved. Students demonstrated an acute awareness of how factors like the physical environment—whether public, private, or professional—and the generational and cultural dynamics between participants shaped the tone and content of their discussions. Situational adaptation, in particular, reflected students’ ability to recognize and respond to the contextual nuances of a conversation, such as the modality (in-person or online), location (public or private), and the nature of their relationship with the other person. Meanwhile, generational and cultural differences added another layer of complexity, as students navigated differences in values, life experiences, and social identities. Together, these adaptations enabled students to manage political conversations more effectively, balancing the need to stay true to their perspectives with the need to respect the particulars of the conversational context.

#### 3.3.1. Situational Adaptation

Situational adaptation emerged as another essential strategy for students navigating politically charged conversations. Unlike strategies that focus solely on content or emotional regulation, situational adaptation reflects the students’ awareness of contextual and relational nuances that shape how they engage with others. As Grace noted, “But when some, like other people are around, the conversation can change,” reflecting how external factors influence the flow of discussions. Similarly, Michael emphasized the importance of context, stating, “I wouldn’t necessarily say it’s appropriate to have [a political conversation] in certain environments.” Students also adjusted based on the dynamics of their surroundings. Nick described these moments, explaining, “Yeah, definitely, like, it’s like right in front of you, so you have to deal with it,” emphasizing the immediacy of some conversations. These examples illustrate the diverse ways students navigated situational challenges to maintain respectful and constructive interactions. While these examples highlight the richness of situational adjustments, the findings show that politically charged conversations are primarily shaped by two factors: conversation modality and location, as well as the nature of the relationship between participants.

*Conversation Modality and Location:* Students frequently tailored their strategies based on whether discussions were in-person, over the phone or online, whether they occurred in public or private settings, or the institutional context in which they found themselves. These distinctions often shaped the tone, openness, and participants’ willingness to engage. Clarissa, for example, reflected on how face-to-face communication shaped the depth of topics discussed: “I feel like when we’re face to face, it’s more like a casual kind of hanging out kind of thing,” suggesting that discussions in more personal and physical spaces tend to avoid tension. Conversations that occur online or over the phone can make it seem easier to take on contentious topics, though it may not always be the most effective. Ashley, for example, noted that “[i]t’s easier to say things online, but I don’t always feel it’s productive.” She points out that “[i]n person, it’s different [because] you can read someone better and adjust.” Likewise, Ella stresses that it is often “easier to talk to [her friend] … on the phone” because “she doesn’t feel as intimidated as me physically being there.” This suggests that the medium of communication can significantly influence how participants navigate political conversations, with some settings allowing for more effective exchanges than others.

Public versus private settings represent another important conversational axis along which these discussions do play out. Evan, for example, shared his reluctance to participate in political conversations in public settings, explaining:

“I feel like if you’re in public, there’s always going to be someone, whether they’re on the left or the right, if they don’t agree with you, … [they] are going to … be obnoxious about it. And it’s just something … I don’t need to deal with. So, it’s like, I’d rather not be in that situation.”—Evan (Male, 24)

Similarly, Hazel noted the potential risks of having intense political discussions in crowded spaces, saying, “If we’re just in a place with a lot of other people, I just don’t think it’s appropriate to have what might become a really intense discussion … You might get loud or anything like that. I think it’s kind of rude.” Rex added to this sentiment, explaining that if he were at a bar, he “probably would not” have political discussions because it would be the “last thing on [his] mind.” These examples underscore how students adapted their behavior to fit the context of public spaces, opting for more caution and restraint to navigate the social risks of openly sharing political opinions. Others, like Mark, emphasized that conversations might be easier in private or semi-private settings. He shared, “I might say things in private, I wouldn’t say in public… I might even go a little bit further with my views in private.” Similarly, Grace and Evan described feeling more comfortable engaging in political conversations at home, where they were less likely to be judged by others. Grace noted that home felt like “a nonjudgment zone… more of a security thing, where you know others won’t hear and judge,” while Evan explained, “I prefer having political conversations at home where it’s safer and more comfortable.” These reflections reveal how privacy often provides a safer space for more open and honest exchanges. Navigating these public–private differences suggests that students often carefully evaluate contextual cues and choose their engagement strategies accordingly.

Many students also adjusted their behavior in response to institutional dynamics, as each setting comes with its own perceived social scripts for engagement. Kerri, for example, described how the workplace often requires restraint and highlighted the expectation of professionalism in such settings. To her, engaging in politically charged discussions could disrupt the workplace atmosphere or harm working relationships. She put it this way:

“I think that work… it’s respectful to not have political arguments… I think that would be negative in a workplace setting.”—Kerri (Female, 21)

Similarly, Michael reflected on the appropriateness of discussing political topics in academic environments, noting, that he “wouldn’t necessarily say it’s appropriate to discuss it like in a … psychology class,” though he would be open to having such discussions in a political science class. Nick added that “high traffic area[s]” where people “set up … political tables,” such as in front of the student union, are also appropriate settings for these conversations. These comments underscore the significance of context in shaping how and where political discussions unfold in institutional spaces, highlighting that the relevance and appropriateness of such conversations often hinge on the subject matter, setting, and audience. Religious institutions, however, presented additional challenges for political conversations, as Patrice explained. She noted that it is “hard to have political conversation within religious groups or churches,” emphasizing the potential for conflict in settings traditionally viewed as spaces for shared values rather than debate. According to her, colleges, for example, offer much better spaces because they can provide “a safe place” for everyone. Taken together, these findings suggest that certain environments may be more conducive to fostering open and productive political dialog than others.

*Closeness of Relationship:* In addition to considering modality and location, students also adapted their strategies based on the closeness of their relationship with the other person. Conversations with family members or close friends were often marked by greater patience and attempts to preserve the relationship, even in the face of disagreement. Clarissa observed, “I think being in a strong relationship like that helps me know when to say less.” This approach reflects a deliberate effort to prioritize harmony over winning an argument, highlighting how trust and mutual understanding in close relationships allowed for a more measured approach. Similarly, Evan described how familiarity and respect shaped his interactions with family members: “When I talk to my parents, I make sure to tone things down.” His comment underscores the strategic adjustments students made to avoid conflict with those they were emotionally close to, especially in hierarchical family dynamics. For others, the intimacy of family bonds created a safe space for honest discussions. Clarissa’s comments further suggest the necessity of trusted relationships for greater openness:

“I really don’t talk about political stuff that much openly unless it’s between, like, super close friends or family,”—Clarissa (Female, 19)

Conversely, interactions with strangers or acquaintances required different strategies. Discussions with strangers often lacked the emotional investment of close relationships, leading to more guarded or detached interactions. Hannah explained, “Talking to strangers about politics feels pointless most of the time,” reflecting a pragmatic approach to minimize unproductive debates with individuals unlikely to share common ground. Similarly, Clarissa shared her reluctance to engage in contentious discussions with unfamiliar audiences, stating, “I definitely wouldn’t be able to agree or speak up if someone was shouting, especially with a stranger. With friends, I know we can disagree, and it won’t hurt our relationship.” For some students, the nature of their relationship also dictated the level of honesty and engagement. Evan shared his preference for discussions with close friends, explaining that “if it comes up,” it’s best to be “someplace that you don’t have to worry about what you say because I mean it’s your close friends, you know … their beliefs” and trust them. This highlights how students carefully weighed the nature of their relationships and negotiated emotionally safe spaces within close friendships to foster more meaningful political discussions.

Through these varied forms of situational adaptation, students adeptly managed political conversations by tailoring their approaches to the context, whether shaped by modality, location, or the nature of their relationships. These strategies highlight their ability to assess and respond to contextual cues, ensuring that conversations remain respectful, productive, and appropriate to the setting. Together, these approaches allowed students to engage thoughtfully while minimizing conflict, reflecting the nuanced ways in which they adapted their communication strategies to suit different conversational dynamics. However, situational adaptation extends further, as students also remained attentive to generational and cultural factors that shaped the tone and openness of political discussions, requiring additional adjustments to navigate differences in norms and expectations effectively.

#### 3.3.2. Generational, Cultural and Subcultural Differences

Generational and cultural dynamics emerged as additional key contextual factors shaping how students approached politically charged conversations. These dynamics highlight how differing social backgrounds and the lived experiences they reflect can further complicate discussions. Generational divides, in particular, reveal how contrasting worldviews between younger and older generations are shaped by traditional norms, life experiences, and media habits, often creating tension around values and priorities. Cultural and subcultural differences added another layer of complexity, as issues tied to race, ethnicity, and systemic power dynamics brought emotionally charged topics to the forefront, challenging students to navigate identity and societal inequalities. These dynamics not only influenced the content of political conversations but also underscored the broader challenges of bridging divides in an increasingly polarized social landscape.

*Adaptation to Generational Differences:* Generational differences emerged as a significant factor influencing how students navigated political conversations, often requiring a balance between respect for older family members and advocating for their own perspectives. These divides frequently reflected differences in values, life experiences, and access to information, making political discussions nuanced and, at times, challenging. Several participants highlighted the difficulties these divides created. John, for example, commented that “talking about homosexual agenda” is easier with his peers than with his “farmer grandparents in North Georgia.” Taylor and Stephen also noted their reluctance to discuss these sensitive issues because they feel like older people “just don’t understand these issues.”

“LGBTQ issues are not something I’m willing to discuss with older people because they grew up in a different society, and it feels like a waste of time.”—Taylor (Male, 22)

“Older people just don’t understand why certain issues are important to us… they grew up in a different world, so they just don’t get it.”—Stephen (Male, 21)

Similarly, Louise described how religion framed her family’s political views: “Generational might probably play a part… my grandma always said you have to vote for the Catholic runner.” These accounts reveal how traditional values often constrained dialogue, especially when older relatives adhered to deeply ingrained ideologies. Other participants pointed to generational differences in media consumption and life experiences as key sources of tension. Samuel explained how older relatives often relied on traditional media, contrasting their beliefs with his own, which were shaped by digital platforms: “Older people… don’t have a political view [from] Instagram or YouTube… my grandma [relies on] Fox News.” Michael added that older generations’ life experiences naturally shaped their priorities, stating, “They come from different childhoods, they come from different lifestyles… that’s what they’re going to support.” Melissa described disagreements with her parents over reform, explaining that while they believed in fixing existing systems, she felt the need to “break things down” and start anew. These examples illustrate how students reconciled conflicting perspectives while striving to keep conversations constructive. Despite these challenges, some participants noted growth in their intergenerational relationships. Lucas reflected on how his relationship with his parents evolved, stating, “Back then… I was always scared I could disagree… nowadays it’s okay. They now know I have different points of view.” Such shifts demonstrate the potential for greater understanding and compromise over time. By acknowledging generational differences and working to find common ground, participants maintained meaningful connections with older family members while expressing their own beliefs. These interactions reveal both the tensions and possibilities inherent in navigating intergenerational divides.

*Cultural and Subcultural Dynamics:* Cultural and subcultural differences emerged as key factors shaping how students navigated political conversations. These differences, rooted in race, ethnicity, cultural backgrounds, and systemic power dynamics, often influenced the tone and direction of discussions. For many participants, these interactions required navigating deeply personal experiences while remaining sensitive to the broader societal implications of these differences. Several participants reflected on the challenges of discussing issues tied to race and cultural identity. Levi described how the demographic composition of his school made it difficult to engage others on topics like police brutality: “When I go to school… it’s very dominantly Caucasian and police brutality isn’t as… personal to a lot of the residents [in this town.]” Similarly, Hazel shared the emotional toll of representing her identity in predominantly white spaces, stating, “Usually I’m… the only Black person in the class. I feel like I have to defend us.” Mark, as a white person, acknowledged this dynamic, reflecting, “I realize that I’m a white man and like neither of those things [women’s issues and racial issues] and have none of those lived experiences.” These accounts highlight the pressure students felt to explain and justify their perspectives in environments where their experiences were often overlooked or misunderstood. Other participants emphasized how national politics intersected with cultural identity. Hannah, reflecting on her adopted sister’s background as a Mexican immigrant, explained how certain political discussions in the current climate can become deeply personal and emotional. She puts it this way:

“She’s from Mexico. But she looks [at] the way Trump has been treating and saying very disgusting things about pretty much her… it’s very emotional for her [to talk about these issues] because of the way she’s been treated here in America.”—Hannah (Female, 21)

Grace also described the challenges of discussing systemic racism within her boyfriend’s family, pointing to different information sources of their views: “His parents are from Colombia… they watch a lot of … cop propaganda and China propaganda” making it difficult to have meaningful “conversations with his parents.” Similarly, Andrew highlighted the personal risk involved in certain discussions, stating, “If I didn’t think that my safety was in danger… I’d be more inclined [to speak]… in front of a white nationalist… maybe not [about] communism … [but still].” These accounts highlight how cultural and subcultural dynamics often complicate political conversations, as differing identities, experiences, and media influences shape perspectives and create misunderstandings.

Unlike conversational strategies related to modality or location, which are often shaped by immediate contextual factors such as the setting or medium of communication, generational, cultural, and subcultural dynamics stem from deeply rooted societal divides. These dynamics are influenced by longstanding differences in values, norms, and social identities, often creating significant tension in political discussions. As a result, participants are compelled to either navigate these potentially fraught discussions with heightened sensitivity and adaptability, carefully considering the perspectives and experiences of their conversation partners or avoid engaging in them altogether to sidestep conflict or discomfort. These divides require students to adjust not only the content of their conversations but also the emotional and relational approaches they take. These dynamics—along with the earlier core themes discussed such as disengagement and negotiation strategies—are situated within and reflective of deeper typification processes, as the next section will explore.

### 3.4. Information Processing

Information processing refers to the cognitive mechanisms that individuals use to organize, interpret, and apply information during political conversations. These mental schemas and heuristics help participants make sense of political discourse and guide their interactions with others. Through these processes, individuals categorize information, form judgments, and decide whether or not to engage in political discussions. In the context of political conversations, information processing serves as a framework that shapes whether individuals will actively engage in dialog or withdraw, depending on how well the new information aligns with their pre-existing beliefs. Participants process information through two interrelated mechanisms: Typification and Information Congruency. Typification involves creating prototypes or mental categories that allow individuals to quickly assess others and decide whether to engage in conversation. Information congruency, on the other hand, highlights how individuals seek out or accept information that supports their pre-existing beliefs, often avoiding conversations that challenge their ideological frameworks. Together, these processes provide a lens through which participants navigate political discourse—before, during, and after the conversation.

#### 3.4.1. Typification Processes

Students often navigate political conversations by falling back on simple typification processes in which more complex ideas are reduced to broad categories based on political beliefs, social identities, and stereotypes. This involves categorizing others quickly, often by their political affiliation, religion, race, or social group, and making assumptions about their views and behaviors based on these affiliations. These cognitive shortcuts allow students to manage interactions more efficiently but often limit the depth of engagement, reinforcing existing divides and making it harder to engage in nuanced or open dialog.

*Typification Based on Political Beliefs and Stereotypes:* Students frequently navigated political conversations by categorizing others based on their political beliefs and stereotypes. This typification process reflects a cognitive strategy aimed at simplifying complex interactions, allowing students to make quick judgments about others’ views and behaviors. However, this approach often reinforced existing divides and limited opportunities for meaningful dialog. Political affiliation frequently served as the primary determinant in categorizing others, with stereotypes acting as shorthand for predicting attitudes and behaviors. For instance, Evan reflected on how his view of liberals was shaped by their association with social actions: “It’s just when you see the masses of liberals rioting and destroying cities… it is hard to not put that tag on the group.” This highlights how visible or polarizing group actions can lead students to quickly categorize others in ways that may not accurately reflect individual beliefs, framing political engagement in terms of “us versus them.” Similarly, John described how he immediately categorizes Republicans into predefined stances: “If somebody’s like, yeah, I’m a Republican, I’m immediately like, OK, well, this is probably how they’re going to feel about this… anti-immigration, pro-life, pro-gun.” This preemptive categorization reflects how political labels simplify complex beliefs, leading students to make assumptions without engaging deeply with the individual’s actual views. It shows that typification can streamline political discussions but also glosses over the diversity within a group. Visual cues, such as political symbols or clothing, further reinforced these categorization processes. Levi expressed this when he said,

“[B]efore I say anything bad about Trump or whenever…I see some guy [wearing a] ‘Make America Great Again’ hat with the big pickup truck and a nice little, huge beard. [I decide ok], let’s avoid politics in all forms of fashion [with this person].”—Levi (Male, 20)

This example highlights how visual symbols—often linked to strong political identities—prompt automatic judgments, influencing decisions on whether to engage or avoid a conversation. Levi’s experience illustrates how typification extends beyond ideological labels to include visual cues, such as political symbols, that shape how students approach social interactions. Similarly, Mark reinforced the idea that typification based on political affiliation shapes expectations about specific issues. His quick assumption about Republicans’ stance on abortion—“if they’re Republican and they’re against, like abortion, I already knew that”—demonstrates how typification simplifies political differences, reducing complex issues to binary categories. The reliance on political labels as indicators of personal views suggests a tendency to engage with people based on expectations rather than genuine understanding. These processes cut across ideological camps and often limit the potential for nuanced, meaningful exchange. Both sides engage in typification, as Taylor elaborates, describing how each camp creates caricatured mental images of the other:

“They paint a picture of a bearded redneck and a trucker cap with an AR15 or for the college ones, a douche bag in a polo shirt with that hair that’s buzzed on the sides standing in the middle of campus shouting at people in the most obnoxious way of looking at it. On the other side, they paint like… the limp dick turtleneck wearing socialists who dye their hair a whole bunch of colors and identify as something absurd.”—Taylor (Male, 22)

*Typification Based on Social Identity:* The tendency to categorize others based on their social identity—such as religion, race, or group affiliation—plays also a crucial role in shaping how students navigate political conversations. Unlike typification tied solely to political beliefs, these identity-driven categorizations further simplify complex issues into binary divisions, shaping assumptions and interactions in more entrenched ways. The tendency to categorize others based on their identity, such as religion, race, or social group affiliation, also plays a significant role in shaping how students engage in political conversations. Unlike simple categorization based on political beliefs and stereotypes, these identity-driven typifications often further simplify complex issues into binary categories. For instance, Samantha shared how her Christian identity led others to assume she should be Republican, illustrating how social identities are frequently tied to political expectations: “I feel like a lot of people personally, I’m Christian. So I think a lot of people think that means I should be Republican.” This assumption shapes how she navigates political discussions, as others automatically expect her to align with conservative views, even when her own beliefs are more nuanced. Similarly, Grace recounted how her initial disagreement with a friend’s “All Lives Matter” stance led her to categorize her friend’s political views, signaling the intersection of personal identity and political beliefs: “She at first said, all lives matter. And I obviously replied and looked like no. I mean, yes, they do. But that’s not the point…” Grace’s reaction reflects how political positions, often tied to social identity, can define the way students engage with each other, especially when those identities conflict. For Lucas, the categorization process was influenced by the political leanings of his friends. He observed how his friends, all Democrats, would likely criticize Republicans:

“All my friends are like Democratic and they are going to end up bashing Republicans.”—Lucas (Male, 20)

This automatic categorization shapes his expectations and frames the political conversation in terms of preconceived group identities, limiting opportunities for more open and meaningful engagement. These identity-based typifications, however, extend beyond political leanings. Samuel highlighted how he avoided political conversations with his girlfriend’s family, assuming their “super Republican” views would create an insurmountable divide:

“When I talk to my girlfriend’s grandparents and uncles… they’re super Republican and I just… I’m not comfortable talking about [politics] because I know they’re not going to agree with me in any way.”—Taylor (Male, 22)

Samuel’s decision to avoid engagement reveals how social identities tied to political beliefs—such as being part of a politically homogenous family—can dictate the willingness to engage in political discourse. These examples highlight how typification based on identity—whether religious, familial, or political—complicates political engagement. By categorizing others based on assumed beliefs linked to social identity, students not only limit their ability to engage in meaningful conversations but also reinforce societal divides, making it more difficult to move beyond simplistic judgments in politically charged environments.

Typification—as the findings show—plays a key role in shaping how students navigate political conversations. This cognitive process simplifies interactions, enabling students to quickly assess whether to engage in or avoid certain discussions. However, this approach often overlooks the complexity of individual views and reinforces existing divides, making meaningful dialog more challenging. By categorizing others based on stereotypes or phenomenological types, students reduce complex issues to binary oppositions, further entrenching polarization. Ultimately, while typification helps students navigate political interactions more efficiently, it limits the potential for nuanced and authentic conversations, hindering productive political discourse and reinforcing societal divides.

#### 3.4.2. Information Congruency

Information congruency is a cognitive process where individuals selectively seek information that aligns with their pre-existing beliefs while dismissing or avoiding information that challenges their worldview. Many students engage in this strategy through selective exposure to information, avoidance of cognitive dissonance, and emotional comfort, all of which reinforce their existing beliefs. While this tendency helps maintain consistency in their views, it may also reduce opportunities to engage with differing perspectives, reinforcing familiar viewpoints over time.

*Selective Exposure to Information:* This process is primarily driven by selective exposure to information that aligns with pre-existing beliefs. This cognitive process involves seeking out and engaging with content that reaffirms one’s political views while avoiding or disregarding information that challenges them. Samuel’s observation highlights this phenomenon: “[A] lot of people have conversations in school. They don’t…know stuff [that] I know, and I don’t…know stuff they know because we’re not…exposed to that. We’re only exposing ourselves to a specific political view or understanding because that’s what we seek out…to support our claim.” Samuel’s insight demonstrates how individuals consciously or unconsciously curate their exposure to information to maintain consistency with their existing beliefs. This process is also evident in the media consumption patterns of students. For example, Mark shared, “I will turn on Fox News just to see what they’re saying, not because I think they have any good points…” Mark’s selective engagement with news outlets illustrates how individuals filter information, only allowing content that fits their ideological framework while dismissing sources that do not. This selective exposure reflects a strong desire to avoid cognitive dissonance, reinforcing their worldview without confronting contradictory ideas. Similarly, Levi noted,

“People my age… they probably kind of just read from what they… identify themselves as… they read it and they believe it.”—Taylor (Male, 22)

This statement underscores how individuals often gravitate towards media sources that validate their political beliefs, reinforcing existing biases and preventing exposure to alternative perspectives. Kerri echoed this sentiment, adding, “I think social media… helps confirmation bias. I think it has a huge part in that,” further emphasizing how platforms specifically designed to personalize content exacerbate selective exposure, creating an environment where individuals are more likely to encounter information that confirms rather than challenges their beliefs.

*Avoidance of Cognitive Dissonance and Emotional Comfort:* The avoidance of cognitive dissonance plays a crucial role in information congruency, as individuals seek emotional comfort by engaging only with viewpoints that affirm their existing beliefs. Levi explained, “Very few people truly want to disagree with people on a consistent basis,” capturing the natural tendency to avoid contentious interactions that could lead to discomfort. This is evident in the experiences of many participants, who express reluctance to engage in conversations that might challenge their core beliefs. For instance, Ella described how engaging with opposing information can feel destabilizing:

“…rock [their] foundation and it makes [them] nervous…[because they] don’t want everything that [they’ve]…built [their] way of thinking [on] to…crumble.”—Ella (Female, 20)

Ella’s statement reflects the emotional difficulty involved in challenging deeply held beliefs, which often discourages individuals from seeking alternative perspectives. Similarly, Grace shared how conversations are easier when they align with her pre-existing views: “…conversations are … easier when it’s with somebody that knows what you’re saying.” This desire for comfort in familiar perspectives further solidifies the tendency to avoid information that challenges one’s worldview. Stephen also highlighted the challenges of engaging in discussions with individuals who hold opposing views, noting, “I think people are getting their information from biased sources, it really hurts their ability to have conversations about both sides instead of just one.” His comment underscores the difficulty in navigating political discourse when information congruency prevents individuals from being exposed to diverse viewpoints, narrowing their understanding and deepening divides.

*Reinforcement of Existing Beliefs and the Echo Chamber Effect:* The tendency to seek out information that aligns with pre-existing beliefs not only simplifies cognitive processing but also reinforces ideological boundaries, creating what some have described as an “echo chamber” effect. This process is often supported by the social validation participants receive when interacting with like-minded peers, strengthening their confidence in their own views while making engagement with opposing perspectives seem less necessary or appealing. While this dynamic may provide a sense of cognitive ease—and at times emotional reassurance—it ultimately limits exposure to alternative perspectives, further entrenching divides. As Taylor reflected, “going back to your side and saying, ‘hey, did you see how right I was? I am so right.’ [It means that] you’re not going to change any minds [on the other side].” Taylor’s statement highlights how individuals not only reaffirm their own beliefs but also rely on validation from like-minded peers, further entrenching their ideological positions. Similarly, Rex explained,

“Yeah, and I think it’s because you had that confirmation. You know. No one is testing you if you’re right or wrong, it’s like, OK, this guy agrees with me, I must be right.”—Rex (Male, 18)

Rex’s comment reflects how the confirmation of one’s beliefs within ideologically homogeneous spaces strengthens the echo chamber effect, as individuals are surrounded by others who share their perspectives. For Samantha, information congruency was evident in her interactions with her mother: “I think definitely with my mom… we do have similar viewpoints… it’s easy to talk to her about these things.” Samantha’s comfort in discussing political matters with someone who shares her views illustrates her preference for engaging with individuals who validate one’s beliefs, making it easier to discuss political topics without the discomfort of disagreement. On the other hand, Melissa pointed out, “I try to focus on… what I believe about the world. And I think that’s just so fundamentally different from conservative values that I don’t think I would ever really move that way,” highlighting how deeply ingrained beliefs create ideological divides that discourage engagement with opposing viewpoints.

In sum, information congruency serves as a cognitive strategy that reinforces existing beliefs by selectively engaging with information that aligns with one’s pre-existing worldview. By filtering out contradictory information, participants limit their exposure to diverse perspectives, hindering the potential for meaningful political discourse. Together with typification processes, which categorize individuals based on political and social identities, these dynamics not only reinforce political divides but also discourage open, constructive conversations across ideological lines. As a result, they make it more challenging to foster productive political dialog and promote mutual understanding.

### 3.5. Summary: A Practical Inventory of Student Strategies[note 3]

So, what does all this tell us? Are these just scattered snapshots of student behavior—or do they reveal something more? We suggest that these four strategies—disengagement, negotiation, context adaptation, and information processing—may reflect more than isolated tactics for avoiding discomfort or saving face. Instead, they appear to represent a shared conversational toolkit that students, across a range of political leanings and social backgrounds, seem to draw on to manage the emotional, social, and cognitive risks of political disagreement. While we cannot claim this toolkit is universal or exhaustive, the recurrence of these strategies across gender lines, ideological camps, and social identities hints at the possibility that these are not merely personal habits or one-off choices, but socially patterned responses—perhaps shaped by broader cultural and institutional forces that influence how political dialog unfolds in students’ everyday lives. Among these strategies, information processing emerged as particularly prevalent and consistently woven into students’ meaning-making. Participants frequently described efforts to sort “fact” from “opinion,” typify opposing viewpoints, and assess whether new information aligned with their existing beliefs. These descriptions suggest a powerful desire to reduce ambiguity and make sense of political disagreement through cognitive work. Yet students’ efforts often seemed to focus less on determining objective truth and more on managing competing claims in ways that felt personally coherent or socially tenable—especially in relationships that mattered to them, such as with friends, family, or trusted peers. Much of this meaning-making appeared motivated not just by a need for cognitive clarity, but by a deeper desire to remain socially acceptable and relationally connected to those closest to them. While students did not explicitly frame these behaviors as identity work, their accounts imply that such strategies may help maintain both internal stability and social belonging in the face of contested political realities.

While some of these strategies may reflect natural cognitive tendencies, the consistency of these patterns across participants—and the ways students described navigating these moments—suggests they are also likely shaped by social learning and environmental reinforcement. In other words, these strategies may reflect adaptive responses—consciously or unconsciously developed—to the conversational risks posed by a polarized and ideologically primed environment. What emerges, then, is not just a random collection of conversational moves, but a window into how students—and perhaps many of us—learn to navigate the everyday pressures created by larger forces of political division, identity salience, and cultural fragmentation. To be clear, these strategies do not appear to resolve political disagreements, nor do they guarantee deeper dialog or ideological change. But they point to something quietly significant: a persistent, if fragile, orientation toward preserving relationships, protecting a sense of “who I am” and “what I stand for,” and making meaning in the face of discomfort and difference. They hint at deeper compensatory identity work—work that feels especially triggered in high-stakes conversational moments involving moral, ethical, or political conflict. Understanding the nature of this identity work may not just be an interesting academic exercise, but a meaningful step toward recognizing the small, often overlooked ways people try to hold themselves—and their relationships—together in divided times. Whether this deeper understanding holds the potential for fostering new or more constructive conversational toolkits remains an open question. But before we turn to what that might mean, we pause to take a closer look at the identity processes that appear to lie beneath these everyday conversational practices—because it is only in that deeper unpacking that the larger significance of these findings begins to come into view.

## 4. Discussion

So, how do we explain these strategies? Why do students, when they find themselves in conversations involving politically charged topics, tend to fall back on these particular ways of navigating the moment? We argue that these strategies reflect students’ efforts to avoid, sidestep, negotiate, or manage the risks of these conversations—risks that are as much, and possibly more, about protecting their sense of self and their social relationships as they are about political disagreement itself. We believe that the findings are best understood through a combined application of Self-Categorization Theory (SCT) and Identity Control Theory (ICT). Seen through these conceptual lenses, the strategies students describe are reflections of deeper identity work. SCT helps explain how—and potentially why—political identities become primed in these conversational situations, leading students to scan their social environments for cues and work to minimize the perceived risks of political disagreement. ICT, in turn, offers a framework for understanding how students respond once political identity becomes salient, through ongoing efforts to manage, protect, or recalibrate their self-meaning through identity verification processes—efforts aimed at maintaining a coherent sense of self in politically charged environments.[note 4]

### 4.1. Priming Political Identities

Self-Categorization Theory (SCT) helps explain how individuals reduce social complexity by organizing themselves and others into social categories, such as race, gender, or political orientation ([60]; [61]; [62]). Building on Social Identity Theory, SCT further specifies that these categories become situationally salient when environmental or conversational cues bring them to the foreground of interaction ([34]). In this study, we use SCT to interpret how students’ strategies reveal the cognitive work involved in recognizing when political identity becomes socially relevant. Two strategies in particular—context adaptation and information processing—offer especially clear examples of this salience detection process. We argue that these strategies, along with others, suggest that political identity may be more readily activated than many other social identities, given their high visibility, social sensitivity, and perceived stakes in today’s polarized climate.

#### 4.1.1. Context Adaptation: Contextual Salience and Anticipatory Identity Management

Our findings suggest that students’ context adaptation strategies offer a clear example of how political identity becomes situationally salient through the interaction of environmental and social cues. According to SCT, different identities become salient when situational contexts make them so. When social context provides cues that make a particular identity feel relevant, contested, or at risk, those identities become more salient ([61]; [34]). In these moments, individuals begin to see themselves and others primarily through the lenses of the identities that are most salient, and less as unique individual people, which SCT describes as depersonalization ([31]).

Participants in our study frequently described scanning their environments and adjusting their conversational behavior in ways that reflect this dynamic process of situational identity salience. For example, students reported withdrawing from political conversations in public settings, such as classrooms, group gatherings, or online spaces, where they anticipated social judgment, reputational risk, or emotional discomfort. One student described these moments as “not the right place to go there,” pointing to the perceived social stakes of being seen as politically divisive or misaligned with the group. Conversely, they described feeling safer to engage politically in trusted spaces, such as with close friends or family members who shared similar values or where the relational stakes felt lower, many of whom may share the same political identity as the student themselves. These patterns reveal how students actively evaluate the “fit” between their identity and the social setting, a core concept in SCT’s explanation of contextual salience ([61]).

This environmental scanning and behavioral adjustment is not merely passive avoidance; it reflects what SCT would characterize as anticipatory identity management. Students appear to preemptively manage their public self-presentation based on the perceived likelihood that political identity might be activated or challenged in a given context. This is particularly evident in descriptions of students changing their communication strategies depending on who was present, whether shifting their tone in mixed political company, avoiding political topics altogether, or selectively engaging when they believed the interaction would be “safe” or “worth it.”

When taken together, these context adaptation strategies reveal that political identity salience is deeply situational, emerging through students’ active sense-making processes as they navigate complex social environments. SCT helps us understand these moments as trigger points where political identity moves from the background to the foreground of social interaction, setting the stage for the identity verification and management work that follows.

#### 4.1.2. Information Processing: Typification and Category Maintenance

In addition to adapting their behavior to situational context, our findings suggest that students engaged in a second layer of identity management: information processing. This strategy reflects the cognitive work students do to make sense of ambiguous or potentially risky political interactions. Drawing on SCT ([61]; [33]), we interpret these patterns as examples of typification, a process in which individuals sort people and cues into familiar social categories to reduce uncertainty and manage social risk.

Students’ descriptions of information processing illustrate how they often read social cues—such as appearance, language, or behavior—as signals of group membership. Rather than exploring these cues in depth, students tended to fit them into pre-existing group-based categories, activating ingroup-outgroup distinctions that shaped whether they approached or avoided political conversation. This kind of cognitive filtering not only allowed students to preemptively assess social safety, but also served to simplify complex social information, reducing cognitive effort in managing politically sensitive interactions.

Importantly, this strategy appears to reinforce category boundaries, aligning with SCT’s concept of category maintenance ([61]; [34]). By mentally translating uncertain or ambiguous information to fit existing ingroup-outgroup frameworks, students maintained a sense of cognitive and social coherence, even if this meant oversimplifying others’ views or closing off opportunities for more nuanced engagement. While this process likely helps manage the emotional and relational risks of political dialog, it also appears to contribute to the reproduction of social and political polarization.

Taken together, students’ information processing strategies reveal how typification and category maintenance operate as cognitive mechanisms of identity priming, reinforcing political identity salience through the mental organization of social information. These processes complement students’ context adaptation strategies by showing how political identity is not only situationally triggered, but also actively maintained through everyday sense-making practices. Both strategies work together to set the stage for the identity verification and corrective processes described in the next section.

#### 4.1.3. Salience of Political Identities

While SCT recognizes that many social identities can become salient depending on situational factors, our findings suggest that political identities may be cognitively primed more readily than other identities in certain conversational contexts. Students in our study consistently described using contextual and informational scanning processes—such as evaluating who was present, what setting they were in, and what cues others provided—to determine whether political identity might become relevant or risky. These cognitive evaluations were not described for other social identities (e.g., being a student, a friend, or a family member) with the same level of anticipation and preemptive categorization.

Self-Categorization Theory helps explain why political identity may be uniquely positioned to become salient in this way. SCT holds that identity salience increases when a category is perceived as distinctive, contextually relevant, or socially consequential ([61]; [34]). Political identity often meets these criteria in student conversations because it is context-sensitive—its relevance shifts depending on the topic, audience, and setting. Students described constantly monitoring conversational dynamics, not just for explicit political content, but for subtle cues that might signal political alignment or opposition. This included interpreting language, tone, nonverbal behaviors, or social affiliations—demonstrating active cognitive filtering and category-based sensemaking.

In this way, political identity appears to operate as a highly responsive cognitive category, one that students activate or suppress based on real-time assessments of conversational fit and social risk. This makes political identity distinct from other, more stable or backgrounded social categories, which students did not report managing with the same level of anticipatory cognitive effort. By showing how students mentally map social dynamics and conversational context to gauge the salience of political identity, our findings extend SCT’s applicability to understanding the real-time cognitive work involved in navigating politically sensitive interactions.

### 4.2. Primed Political Identities

Once political identity is activated or brought to the foreground, it not only becomes more socially and psychologically relevant but also appears to trigger a series of protective responses aimed at managing that identity in the face of perceived social misalignment or threat. In this sense, priming is not a static state of awareness, but the starting point of a dynamic process of identity management. According to Identity Control Theory ([14]), individuals strive to maintain identity stability by aligning external feedback with their internal identity standards. This process operates through a feedback loop, where individuals continuously compare how they are perceived or received by others with the meanings they associate with their own identity. When feedback misaligns with these internal standards, individuals experience identity disruptions or threats, which can generate emotional responses such as discomfort, frustration, or cognitive dissonance ([14]; [12]; [63]). These moments of misalignment are not simply experienced passively; they often signal the need for corrective action to primarily restore alignment and reestablish cognitive coherence and social balance—and, in some cases, emotional relief. This feedback loop is dynamic and self-regulating, as individuals adjust their behavior, perceptions, or social interactions to achieve identity verification—that is, a state where external feedback affirms internal identity meanings.

The significance of a primed or threatened identity—such as deeply held political or moral beliefs—may shape the intensity and type of corrective action that follows. Based on our data, we suggest that the four strategies identified in this study—disengagement, negotiation, context adaptation, and information processing—can be understood as surface-level reflections of this deeper identity verification process. Whether by avoiding risk (disengagement), modifying approach (negotiation and context adaptation), or reinforcing one’s cognitive framework (information processing), these strategies appear to function as practical, socially patterned responses to the identity challenges posed by politically charged conversations. While students may not have explicitly named these strategies as identity work, their descriptions suggest that such behaviors serve as adaptive efforts to manage self-presentation, protect relational ties, and maintain a sense of internal coherence in the face of social and political tension. In the sections that follow, we unpack these identity verification processes in more detail, highlighting how they can be understood as practical expressions of broader identity management efforts to avoid, modify, or reinforce one’s position when navigating politically charged conversations.

#### 4.2.1. Avoidance Mechanisms

Avoidance mechanisms—through an ICT framework—represent conscious or subconscious strategies individuals use to protect identity standards when faced with potential or actual disruptions in contentious political interactions. ICT emphasizes the importance of maintaining alignment between internal identity meanings and external feedback ([11]). When misalignment occurs—threatening identity verification—avoidance serves as a corrective action to minimize dissonance and preserve identity stability ([14]). The findings of this study reveal that avoidance mechanisms include disengagement processes (e.g., the “time and energy excuse,” keeping the peace, conversation enders) and aspects of context adaptation in politically charged settings.

Avoidance often begins as a preemptive strategy to prevent identity disruptions before they occur. According to ICT, individuals continuously monitor interactions for signals of potential identity misalignment. In contentious political conversations, this leads to disengagement when the likelihood of achieving identity verification is low, such as with someone perceived as unpersuadable. This aligns with the “time and energy excuse”, where individuals withdraw to avoid futile, emotionally taxing interactions. [57] ([57]) supports this finding, arguing that people disengage when conversations pose an emotional or cognitive burden, particularly in highly polarized contexts. Such avoidance protects identity stability by minimizing cognitive strain and potentially preventing the emotional frustration or dissonance that arises from repeated misalignment ([55]). Similarly, [64] ([64]) found that prior negative conversational experiences drive individuals to preemptively avoid future discourse as a means of self-preservation. In this sense, preemptive avoidance functions as a self-protective mechanism, maintaining a coherent sense of self across different times and contexts.

In relational contexts, avoidance takes on the added role of balancing multiple identity standards. Individuals must reconcile their relational identities (e.g., being a good family member) with their ideological identities (e.g., holding strong political beliefs). When political conversations threaten to destabilize this balance, avoidance—through silence or neutrality—becomes a necessary corrective action. The theme “keeping the peace” reflects this dynamic, as individuals prioritize relational harmony over ideological confrontation. This aligns with findings by [21] ([21]), who emphasize that relational harmony often takes precedence in close networks, particularly when disagreement risks long-term relational damage. By recalibrating their behavior to align with contextually dominant identities, individuals preserve stability across multiple aspects of their self-concept. This relational avoidance also reflects the role of strategic self-regulation in ICT: disengaging from conflict helps manage interactional risks and preserves relational and ideological stability over time.

While preemptive avoidance mitigates identity threats before they emerge, reactive avoidance addresses disruptions in real time. ICT highlights that individuals operate within a feedback loop, comparing external feedback with identity standards and adjusting behavior to restore alignment. When political conversations escalate, individuals employ strategies such as “conversation enders”—verbal or non-verbal cues that allow for a graceful exit. Such reactive strategies echo findings by [44] ([44]), who observed that individuals often disengage to prevent escalating tensions in polarized conversations. Reactive avoidance serves two key functions: it prevents further identity misalignment and restores the individual’s sense of control over the feedback loop, reinforcing agency and potentially contributing to the emotional stability of the self.

Taken together, avoidance mechanisms—whether preemptive (e.g., the “time and energy excuse”), relational (e.g., “keeping the peace”), or reactive (e.g., “conversation enders”)—function as deliberate corrective actions within ICT’s identity verification process. By prioritizing stability and minimizing prolonged disruption, avoidance enables individuals to maintain identity coherence and possibly emotional equilibrium across varying interactional contexts. The themes identified in this study provide empirical evidence of how avoidance operates as a recalibration strategy to safeguard identity standards during contentious political conversations. This aligns with broader work in political discourse, where minimizing identity threats is critical for navigating polarized environments ([57]; [64]; [44]).

#### 4.2.2. Modification Mechanisms

Modification processes—through the lens of Identity Control Theory (ICT)—represent strategies individuals use to adjust their behavior or communication to restore alignment between internal identity standards and external feedback during contentious political interactions. Unlike avoidance mechanisms, modification involves remaining engaged while actively recalibrating interactions to minimize identity threats and maintain cognitive coherence and likely emotional stability. The findings highlight key modification processes such as active listening, agreeing to disagree, finding common ground, and context adaptation strategies like situational adjustment and generational alignment as corrective actions to manage identity disruptions. Within ICT’s feedback loop, individuals compare external perceptions with internal identity standards. Identity misalignment—when feedback challenges or undermines identity meanings—requires corrective action to restore balance. Unlike avoidance, modification processes maintain relational engagement while regulating emotional arousal and cognitive dissonance, providing a dynamic means for identity stabilization.

Active listening reflects a key strategy for diffusing identity threats. By prioritizing understanding over immediate defense, students signaled openness, de-escalating ideological tension while preserving their self-concept as respectful or open-minded. ICT highlights that identity verification occurs not only through assertion but through relational behaviors aligning external perceptions with internal identity meanings. Active listening thus may serve as a relational management tool, helping individuals maintain constructive engagement while reducing potential tension. [21] ([21]) argue that active listening fosters mutual respect, minimizing relational tension, while [57] ([57]) shows that it mitigates emotional burdens by reducing perceived hostility.

Another significant modification process was agreeing to disagree, where students acknowledged differences to prevent further misalignment. ICT explains this as a recalibration that protects both ideological and relational identities. By prioritizing internal standards such as tolerance or civility, individuals avoid escalating cognitive discomfort, emotional arousal and relational strain. This stabilization within the feedback loop enables individuals to preserve their self-concept while diffusing immediate threats. While [57]’s ([57]) primary focus is on how self-conscious emotions like pride or shame motivate conformity to group norms, she also acknowledges the role of cognitive processes—such as identity-linked evaluations of information—in shaping political behavior. Taken together, her work suggests that de-escalation strategies may help preserve both emotional stability and cognitive coherence by reducing the strain of prolonged ideological conflict.

Participants also engaged in finding common ground by reframing conversations to emphasize shared values or experiences. This modification process aligns with ICT’s concept of contextual recalibration, where individuals shift their focus to reducing identity misalignment. By highlighting areas of agreement, individuals maintain identity verification while fostering mutual understanding. [45] ([45]) similarly found that exposure to cross-cutting discussion networks can promote political tolerance and reduce social tension, in part by encouraging individuals to focus on shared experiences and relational considerations. However, she also warns that such exposure may dampen political engagement by increasing ambivalence and reducing confidence in one’s views.

In addition, the findings illustrate how context adaptation aligns with modification processes through situational adjustment and generational alignment. Situational adjustment, such as softening tone or shifting engagement style in public versus private settings, reflects a dynamic recalibration to minimize identity threats while managing external perceptions ([45]; [21]). Public settings heighten reputational risks, prompting individuals to adapt behavior to maintain relational and ideological stability, while private spaces allow for more open communication. Similarly, generational alignment occurs when individuals adjust their approach to meeting relational expectations, such as emphasizing harmony with older family members. Research by [66] ([66]) shows younger individuals adapt tone to avoid intergenerational conflict, aligning with ICT’s emphasis on recalibrating feedback loops to preserve relational bonds without compromising self-concept. These adaptive strategies highlight the context-sensitive flexibility of modification processes, enabling individuals to reduce identity disruptions while balancing emotional stability, relational harmony, and ideological integrity ([57]).

Taken together, modification processes—through active listening, agreeing to disagree, finding common ground, and context adaptation—demonstrate how individuals recalibrate interactions to protect identity standards while remaining engaged in contentious political conversations. By dynamically adjusting their behavior, individuals work to restore identity coherence and maintain continuity across shifting social contexts, balancing ideological integrity with relational harmony. While these adjustments may also reduce emotional strain or tension in the moment, our findings primarily highlight the cognitive and interactional work involved in sustaining a stable sense of self. These findings extend ICT by emphasizing the flexible, context-sensitive strategies individuals use to preserve identity stability in politically and socially charged interactions ([11]; [14]).

#### 4.2.3. Reinforcement Mechanisms

Reinforcement processes—through the lens of Identity Control Theory (ICT)—refer to strategies individuals use to affirm and stabilize their identity standards when external feedback poses a perceived threat or misalignment. Unlike avoidance or modification, reinforcement mechanisms strengthen internal identity meanings, often by filtering, prioritizing, or amplifying identity-consistent information. The findings of this study highlight key reinforcement strategies such as typification, information congruency, and leveraging generational and cultural differences as tools to protect and reaffirm both ideological and relational identities in contentious political interactions. Within ICT’s feedback loop, reinforcement strategies function as stabilizing corrections that proactively or reactively manage identity disruptions. [14] ([14]) argue that identity stability occurs when individuals actively align external inputs with internal identity standards, reducing cognitive dissonance and emotional strain. ICT posits that reinforcement mechanisms allow individuals to affirm the coherence of their self-concept by selectively filtering and prioritizing feedback that aligns with their identity meanings.

Typification emerged as a key reinforcement process, where participants categorized others based on perceived ideological or behavioral cues. By grouping individuals into typified categories such as “liberal,” “conservative,” or “uninformed,” participants simplified interactions and reduced perceived threats to their identity. Typification acts as a cognitive filter, allowing individuals to dismiss challenging perspectives while maintaining ideological integrity. From an ICT perspective, this strategy preserves alignment between external feedback and internal identity standards, stabilizing the self-concept. This aligns with findings by [45] ([45]), who observed that individuals rely on stereotypes to anticipate conflict and protect their self-concept in polarized environments. Typification, as [12] ([12]) suggests, provides individuals with a sense of control over their identity verification process by preemptively managing how they interpret external inputs.

Participants also employed information congruency to reinforce identity verification by seeking out identity-affirming information and dismissing dissonant content. ICT highlights that successful identity stabilization occurs when external feedback aligns with internal standards. In this study, participants filtered conversations, media, and interpersonal feedback to prioritize identity-consistent narratives while avoiding content that disrupted their ideological or relational equilibrium. This process aligns with [57] ([57]), who argues that individuals gravitate toward supportive information to reduce cognitive strain and emotional discomfort during ideological conflict. Similarly, [59] ([59]) found that motivated reasoning drives individuals to prioritize evidence that validates pre-existing beliefs, reinforcing their identity while dismissing opposing viewpoints.

Leveraging generational and cultural differences also seemed to act as a reinforcement mechanism, where individuals drew on relational norms or cultural expectations to stabilize their self-concept. For example, participants emphasized harmony or deference when engaging with older family members, framing generational alignment as a way to preserve relational identities while maintaining ideological integrity. From an ICT perspective, this strategy allows individuals to control external feedback by positioning themselves within relationally appropriate frames, reducing potential identity misalignment. This aligns with [66] ([66]), who found that intergenerational communication often involves adjustments to protect relational harmony and maintain identity consistency across cultural or generational divides. Such efforts reflect a dual function of reinforcement: stabilizing ideological identity while safeguarding relational bonds, particularly in culturally sensitive interactions.

Taken together, reinforcement processes—including typification, information congruency, and leveraging generational and cultural differences—demonstrate how individuals affirm their identity standards in politically charged conversations. By filtering external feedback, prioritizing supportive narratives, and relying on relational or cultural norms, individuals protect alignment within ICT’s feedback loop while minimizing cognitive dissonance and emotional disruptions. These findings expand ICT by illustrating reinforcement as a critical stabilizing mechanism, highlighting how individuals actively manage identity coherence amid the relational and ideological complexities of contentious interactions ([11]; [14]).

### 4.3. The Bigger Picture

Taken together, this identity work —avoidance, modification, and reinforcement—provides important insights into how individuals cognitively manage identity threats or identity disruptions in politically and socially charged conversations. By applying Self-Categorization Theory (SCT) and Identity Control Theory (ICT), we show how these recalibration strategies reflect the sense-making work individuals do to anticipate social risk, evaluate conversational dynamics, and adjust their engagement in ways that protect their self-concept when political identity becomes or remains salient. While these strategies seem to carry emotional undercurrents or consequences, our findings suggest that students primarily approach these moments as cognitive challenges—requiring them to interpret social cues, classify interactional risks, and navigate competing identity demands in real time. This highlights the role of cognitive filtering, categorization, and behavioral adaptation as core mechanisms through which individuals manage identity salience and social exposure in politically sensitive interactions. In this way, our study extends ICT scholarship by showing how conversational strategies like disengagement, negotiation, context adaptation, and information processing function as identity work not mere conversational techniques, allowing individuals to preserve cognitive coherence and maintain control in the face of potential identity threats.

## 5. Conclusions

In this concluding section, we reflect on what our findings reveal about how students navigate political disagreement, and consider what these insights might mean for researchers, educators, and practitioners working to foster more constructive dialogue across divides. We begin by summarizing our key findings in relation to the study’s two guiding research questions: how students navigate political conversations across ideological divides (RQ1), and the contextual and relational factors that shape their willingness to engage (RQ2). We then explore why we believe these patterns are best understood as a form of identity work. From there, we turn to the deeper challenge of teaching for identity discomfort—a skill we argue is essential but often overlooked. Finally, we close by reflecting on the potential for reconnecting with the common bonds that unite us, even in divided times.

### 5.1. Identity Work Beneath Everyday Political Conversations

This study sets out to explore how students navigate political conversations across ideological divides—especially why they sometimes avoid these conversations and under what conditions they feel more open to engaging. Through interviews with 30 students from diverse social and political backgrounds, we identified four core strategies students use to manage these moments: disengagement, negotiation, context adaptation, and information processing. These strategies reflect not just surface-level behaviors but appear to be grounded in deeper identity management processes. Seen through the combined lenses of Self-Categorization Theory and Identity Control Theory, they seem to function as adaptive responses to the social and cognitive risks that arise when political identity becomes salient in conversation. Students’ descriptions suggest both direct and indirect efforts to protect their sense of self, maintain valued relationships, and make sense of competing claims in ways that feel personally coherent and socially tenable.

In this way, our findings offer a more grounded understanding of the everyday identity work that occurs when students face politically charged conversations. This expands existing literature by showing that even seemingly simple conversational moves—such as changing the subject, agreeing to disagree, or seeking common ground—are not merely communication “tactics”, but meaningful forms of conscious or subconscious identity work aimed at solving a familiar social problem: how to protect both the self and one’s relationships in conversational situations that feel risky or high-stakes. These adaptive responses appear to be shaped by larger social forces unfolding in students’ communities and institutions—forces reinforced by social norms about what is safe or appropriate to talk about and shaped by lifelong social learning about how to manage disagreement, particularly around political issues. In this sense, the strategies students describe reflect not only personal coping mechanisms but also socially patterned ways of navigating the political tensions that increasingly define everyday life.

What is particularly striking is that these strategies seem to cut across political identities, social groups, and demographic categories. Regardless of where students positioned themselves ideologically, they described drawing on remarkably similar conversational techniques to manage the discomfort of political disagreement. This consistency suggests that these are not just individual coping styles but socially patterned and culturally reinforced ways of navigating political tension. In other words, they reflect a shared cultural playbook—one that students seem to learn, adapt, and perform in response to the relational risks of engaging across political lines.

At the same time, these strategies present a double-edged sword. On one hand, they help students maintain social harmony, avoid conflict, and protect relationships in the moment. On the other hand, they may also limit opportunities for deeper dialog or ideological learning. By normalizing avoidance, surface-level agreement, or strategic withdrawal, these strategies may reinforce long-term patterns of disengagement and polarization. While they are adaptive in the short term, they risk making political difference feel unmanageable in the long term by encouraging people to sidestep the very conversations that might open pathways to understanding or change.

### 5.2. Learning to Live with Identity Discomfort

This tension raises important implications for how we think about the work of dialog. Most interventions aimed at improving political discourse tend to focus on cognitive or rhetorical skills—such as how to make better arguments, listen more effectively, or find common ground. Yet our findings suggest that the deeper challenge may be less about skill deficits and more about how people manage the identity discomfort that such conversations often trigger. Navigating political difference, it seems, is not just an intellectual task but a social and emotional one—a process of managing the risks to self and relationship that come with speaking, and listening, across divides.

If, as our findings suggest, students use these conversational strategies as identity protection mechanisms, then what may be most urgently needed is not just training in argumentation or civility, but the development of greater tolerance for identity discomfort itself. This may require a shift in how learning environments and dialog interventions are designed. Rather than treating discomfort as something to avoid or quickly resolve, it may be more productive to normalize discomfort as part of the learning process—a view supported by research in transformative learning ([43]), which highlights the potential value of disorienting dilemmas in promoting growth.

There are already models that seem to move in this direction. Intergroup dialog programs ([67]), for example, offer structured spaces where participants are encouraged not just to share perspectives, but to sit with the discomfort of identity tension in a supported, relationally accountable way. These programs seem to emphasize the development of emotional and relational capacities—such as empathy, self-reflection, and sustained engagement—as complements to more traditional cognitive skills like argument analysis or fact-checking. Likewise, research on moral psychology and political identity threat ([29]; [10]) suggests that moral and political disagreements are often experienced as existential threats, not simply intellectual disagreements—implying that meaningful engagement may require helping people feel safe enough to risk being wrong, misunderstood, or changed.

In this light, we argue that the future of dialogue practice may hinge on our ability to help students build a tolerance for identity discomfort—to prepare them not just to tolerate, but to work with the tension that arises when their core identities are challenged. This might include creating “brave spaces” ([2]) that balance care and accountability, offering tools to help students recognize and manage their own identity stakes, and fostering relational trust that makes it more possible to stay engaged when conversations get hard. While we cannot yet offer a definitive model for how to do this, our findings suggest that naming and exploring the identity work beneath everyday conversational strategies may be an important place to start. By shifting attention from surface-level communication tactics to the deeper social and emotional processes that shape how people engage across divides, we may begin to open up new possibilities for dialog that is not only more honest, but perhaps also more sustainable in the long run. Of course, recognizing the role of identity discomfort avoidance is only the beginning. The harder question is what comes next. If, as we have argued, these conversational strategies are socially learned and reinforced over time, there is reason to believe they can also be relearned or reshaped. This is where we believe both the research community and the educational community have important work to do.

For researchers, our findings open the door to new lines of inquiry focused on how people learn to manage identity discomfort, and whether alternative conversational strategies—ones that move beyond avoidance toward more meaningful engagement—can be taught and sustained. This includes testing interventions that help individuals recognize when their identities feel challenged and develop skills to stay engaged despite discomfort. It also includes exploring how different social contexts—such as classrooms, workplaces, or online spaces—either reinforce or disrupt the avoidance patterns we have described. While we cannot yet specify what an ideal conversational “toolkit” might look like, the search for such a toolkit feels both urgent and possible.

For educators and practitioners, there are already promising practices that seem to point the way. Initiatives like the Human Library ([51]), where people “check out” human beings to hear their personal stories, and Listening Projects (e.g., [19]) that bring community members together across divides, offer creative approaches to fostering understanding and humanizing political “others.” Similar outcomes have been observed in large-scale deliberative experiments like the “America in One Room” project, where structured, face-to-face discussions among politically diverse citizens not only increased mutual understanding but also reduced polarization—effects that persisted over time ([24]). While these programs do not eliminate identity discomfort, they create structured spaces where people can face that discomfort with support and accountability—an approach consistent with [1]’s ([1]) Contact Hypothesis and [49]’s ([49]) reformulation, which emphasizes that cross-group contact is most effective when it involves sustained interaction, equal status, and cooperative goals.

Closer to home, we believe there is also important work to do in everyday spaces—around kitchen tables, in classrooms, and in workplace conversations. These are the spaces where people have regular opportunities to practice staying engaged when things get uncomfortable, to listen with curiosity rather than judgment, and to resist the impulse to shut down or walk away. Yet many of us—including students—may feel unprepared for this kind of engagement, in part because formal education often prioritizes debate and critique over connection and collaboration. As educators ourselves, we are particularly concerned that much of what students learn in K-12 and higher education may unintentionally reinforce division rather than prepare them for meaningful dialogue. We often teach students how to win arguments, critique ideas, and deconstruct opposing views—all valuable academic skills, but ones that can train students to treat conversations as zero-sum games, where someone must win and someone must lose. What is often missing is the relational work of finding common ground, identifying shared values, and building trust across differences. These are not innate capacities; they are learned skills that require intentional practice, relational support, and social reinforcement ([19]; [67]).

Again, we do not pretend to have all the answers, but we hope this study helps reframe the conversation about what it means to prepare people—especially students—for political engagement in divided times. By naming and unpacking the identity work that underlies everyday conversational strategies, we hope to invite scholars, educators, and practitioners to continue this work, to build on these insights, and to imagine new ways of helping people move beyond avoidance toward more honest, courageous, and relationally grounded engagement across divides.

### 5.3. Rediscovering What Unites Us

Let us push this a step further and draw on the excellent work of political scientist Matthew Levendusky, who in his recent book Our Common Bonds ([38]) reinforces this call to rethink how we engage across divides. Levendusky’s research reminds us that while affective and ideological polarization are real and well-documented, they do not erase the many other identities that unite us. We are not only partisans or ideological actors—we are also family members, friends, neighbors, sports fans, coworkers, community members and fellow country “men”. These identities, too, are deeply meaningful and socially salient. Levendusky shows that when these common identities are primed, people become more open to seeing those on “the other side” not as enemies, but as fellow human beings with whom they already share meaningful connections.

This, we believe, brings the work of identity discomfort full circle. If people can learn to sit with the discomfort that comes when their core political identities feel challenged, and if we can help them also remember and activate the many other identities they share with those across the aisle, there may be a path forward. This will not eliminate difference, nor should it. Democracy depends on difference—on having competing ideas, values, and visions for the future. But what seems increasingly at risk is our ability to work through those differences without giving up on one another in the process.

Reconnecting with our common bonds, while learning to tolerate and work through identity discomfort, may offer a fragile but real way to begin that work. It is not a quick fix, nor is it a guarantee. But if we can help students—and all of us—see difference not as a threat to be avoided, but as a strength to be navigated, we may take one small step toward rebuilding the social and relational fabric that polarization so often threatens to tear apart. And if that work happens not just in classrooms or dialog programs, but in the everyday spaces where people live, work, and build community together, then perhaps we can begin to reclaim the possibility of constructive engagement—one imperfect, necessary conversation at a time—conversations that, while small, may just hold the seeds of something larger.

## 6. Limitations and Directions for Future Research

While our findings offer valuable insights into how students navigate political disagreement, they are based on a small, qualitative sample, which limits the generalizability of the results. This underscores the need for future research using larger, more diverse populations to test and extend these insights across broader contexts. Although we did not observe significant variation in the strategies students described across political or demographic groups, we recognize that our sample may not have captured the full range of identity dynamics at play. Future studies could explore how these processes unfold across different ideological affiliations, cultural or racial identities, and types of relationships (e.g., peer-to-peer vs. intergenerational interactions). Additionally, it may be useful to examine how different conversational settings—such as online versus face-to-face exchanges, or public versus private spaces—shape the emotional and identity-related stakes of political dialog. Understanding these contextual dynamics could further refine interventions aimed at helping people move beyond avoidance toward more sustainable engagement across divides.

Beyond mapping strategy use across settings and identities, future research should consider examining the deeper identity work that underlies these conversational strategies. The four strategies identified in this study—disengagement, negotiation, context adaptation, and information processing—should not be seen as isolated techniques, but as surface-level expressions of ongoing identity regulation. Building on our findings and theoretical synthesis, we propose that three interrelated identity processes—identity salience, identity coherence, and feedback management—may serve as a useful framework for future inquiry. Research might explore how conversational cues or social contexts prime particular identity positions, how individuals work to sustain coherence across moments of dissonance or threat, and how they interpret and respond to perceived recognition or misrecognition by others. This line of inquiry could help bridge political discourse research with broader theories of self-concept maintenance and identity regulation, while also clarifying how these processes may vary across relational dynamics, ideological intensities, and institutional settings such as classrooms, families, or digital platforms where self-presentation carries distinct forms of risk and accountability.

We also point to the need for research that examines whether interventions aimed at building identity discomfort tolerance can shift these patterns over time. Our findings suggest that many of the strategies students employed were oriented toward avoiding discomfort—through deflection, withdrawal, or over-correction—often at the cost of deeper engagement. While these approaches may offer short-term benefits, they may also limit opportunities for reflection, learning, and mutual recognition. Future work could explore whether alternative forms of identity work—ones that accept temporary incoherence, ambiguity, or friction—might support more constructive and relationally sustainable political conversations. This includes asking what conditions make such discomfort tolerable rather than overwhelming, and what kinds of support—pedagogical, relational, or institutional—can help individuals remain open when their identities are challenged. Moving beyond avoidance-versus-engagement dichotomies, future research might develop a more nuanced understanding of identity work as a developmental process. If political dialog is, at its core, a space where people try to maintain integrity while staying in relationships, then the ability to stay with discomfort may be a core civic skill. Interventions that foster identity flexibility, narrative complexity, or dialogic resilience—whether in educational contexts, peer facilitation, or public deliberation—could offer powerful starting points for cultivating that capacity.

While our focus has primarily been on the cognitive and identity management aspects of these strategies, our findings also suggest that emotional dynamics—though less consistently foregrounded—may play an important reinforcing or motivating role. In particular, students’ choices to disengage, adapt, or recalibrate often seemed driven not just by calculated self-presentation, but by attempts to manage affective discomfort—such as anxiety, shame, or the fear of social rupture. Future research could build on this by more explicitly examining how identity threats are not only cognitive disruptions but also experienced as emotional or value-laden challenges to the self. Work by [22] ([22], [23]), for example, positions values as core components of identity, suggesting that threats to deeply held values may be experienced as existential challenges to one’s sense of self. This emotional dimension may be especially acute in conversations where perceived value conflict overlaps with fears of exclusion, misrecognition, or moral condemnation. We also see potential for research that explores how people manage these affective dynamics in real time—whether through suppression, redirection, or relational repair. Emotional regulation strategies, especially those tied to identity protection, may be just as central as cognitive ones in determining whether political conversations break down or move forward. Exploring this intersection between values, emotion, and identity management could provide a richer understanding of the affective stakes of political conversation—and illuminate what it takes not just to stay cognitively coherent, but emotionally present, in moments of political tension.

In short, while this study begins to map the contours of how people navigate political difference, it also opens space for a broader research agenda—one that looks beyond conversational tactics to the deeper identity negotiations they reflect and asks what it might take to support individuals in staying present with discomfort, rather than retreating from it. We hope our findings provide a useful starting point for scholars and practitioners who wish to take that work further.

## Figures and Tables

**Figure 1 behavsci-15-00835-f001:**
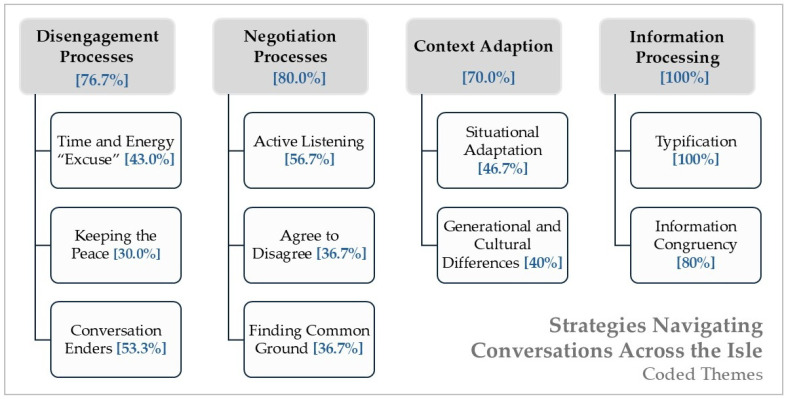
Researcher-coded strategies students use to navigate political conversations across the aisle. Note: Percentages in brackets indicate the proportion of participants who used each strategy. See the Methods section for further explanation of how these coverage rates were determined and their interpretive meaning.

**Table 1 behavsci-15-00835-t001:** Participant characteristics with interview times.

#	Pseudonym	Politics	Gender	Age	Race	I-Time *
1	Ashley	conservative	female	20	Caucasian	66:47 mins
2	Ella	conservative	female	20	Caucasian	54:52 mins
3	Tiffany	conservative	female	19	Caucasian	75:05 mins
4	Clarissa	conservative	female	19	Caucasian	50:46 mins
5	Kerri	conservative	female	18	Caucasian	39:03 mins
6	Evan	conservative	male	24	Caucasian	63:12 mins
7	Edward	conservative	male	**	Caucasian	25:50 mins
8	Nick	conservative	male	19	Caucasian	55:50 mins
9	Rex	conservative	male	18	Caucasian	53:45 mins
10	Zack	conservative	male	21	Caucasian	55:24 mins
11	Taylor	liberal	male	22	Caucasian	74:25 mins
12	Brittany	liberal	female	18	African American	73:56 mins
13	Tammy	liberal	female	21	Multiracial	67:29 mins
14	Louise	liberal	female	22	Caucasian	62:59 mins
15	Alyssa	liberal	female	19	Hispanic/Latino	68:08 mins
16	Hazel	liberal	female	20	African American	37:00 mins
17	Stephen	liberal	male	21	Hispanic/Latino	52:15 mins
18	Levi	liberal	male	20	African American	86:24 mins
19	Mark	liberal	male	22	Caucasian	73:31 mins
20	Tony	liberal	male	22	Caucasian	33:19 mins
21	Grace	other	female	23	Caucasian	59:52 mins
22	Patrice	other	female	19	Caucasian	49:37 mins
23	Melissa	other	female	20	Caucasian	65:57 mins
24	Samantha	other	female	18	Caucasian	55:06 mins
25	Andrew	other	male	24	Hispanic/Latino	43:49 mins
26	Lucas	other	male	20	Hispanic/Latino	48:23 mins
27	John	other	male	22	Caucasian	66:16 mins
28	Samuel	other	male	18	Caucasian	78:55 mins
29	Michael	other	male	20	Caucasian	39:53 mins
30	Hannah	other	female	21	Hispanic/Latino	54:46 mins

Notes: * I-Time … interview time, ** … missing age of participant.

## Data Availability

The dataset generated and analyzed during this investigation is not publicly available to protect the privacy of the participants. However, portions of the dataset may be available from the corresponding author upon reasonable request.

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
