# Peer review of "When Politics Gets Personal: Students’ Conversational Strategies as Everyday Identity Work"

_behavsci, 2025, doi:10.3390/bs15060835_

Round 1

Reviewer 1 Report

Comments and Suggestions for Authors

The authors identified college students’ strategies for disengaging political conversation with opposite ideology supporters. Using semi-structured interviews with 30 students, the authors identified 4 key strategies that students use in political conversations: disengagement, negotiation, adapting to context, and processing information. Although I found this topic interesting, I have some major concerns. These are addressed in more detail below.

  • My major concern is related to discrepancy between research questions and findings. The main research questions the authors intended to address were to figure out “why people are avoiding political conversations”, and “under what conditions people may be more open to having political conversations with ideologically dissimilar others”. However, I do not see answers for these two research questions.
  • The authors argue that students may avoid political conservation with ideologically dissimilar others as a form of protecting their social identity. In this argument, emotions and social identity were proposed as key conceptual background. However, no empirical evidence was provided to support this claim.
  • The authors provided emotional explanations for avoiding political conversations as discussing political issues generates negative emotions (stress and anxiety); however, this link has not been tested. Also, the four strategies people use in political conversation do not seem to provide answers to the main research questions that authors discussed in the manuscript.
  • Is this effect domain specific (political identity) or domain general (social identity in general)? The authors claimed that people tend to disengage as a means of preserving self-consistency and emotional equilibrium. Wouldn’t this be the case for any other identity?
  • Basically, I do not see a connection between the results the authors provided and the major research questions the authors proposed. Also, it might be helpful if the authors provide more convincing reasons why the findings are important. For example, how do the four strategies relate to identity control theory?
  • The interview questions do not seem to address social identity issues. As the authors base their research on social identity theory, related literature needs to be discussed more in depth and results need to be discussed related to prior findings.

Author Response

Issue 1: Research, Methods, Findings & Theoretical Framework

"My major concern is related to discrepancy between research questions and findings. The main research questions the authors intended to address were to figure out 'why people are avoiding political conversations', and 'under what conditions people may be more open to having political conversations with ideologically dissimilar others'. However, I do not see answers for these two research questions."

Our Response

We sincerely thank Reviewer 1 for this critical and constructive push. Your concern helped us recognize a key shortcoming in the earlier version of our manuscript: the lack of a fully integrated narrative that clearly linked our research questions, findings, and theoretical framework. This observation prompted us to step back and strengthen the theoretical and methodological coherence of our work.

We now see that the disconnect you flagged was not just a matter of how we presented the manuscript, but a deeper issue of how we framed the logic of our qualitative approach. While our original questions aimed to explore why students disengage from or engage in political conversations, our initial framing mistakenly suggested a more deductive, hypothesis-testing design. In reality, this was an exploratory, interpretive study grounded in student meaning-making, consistent with constructivist traditions that prioritize understanding over prediction (Braun & Clarke, 2006; Charmaz, 2014; Berger & Luckmann, 1966).

In response to your valuable critique, we have made several key revisions throughout the manuscript to better align our research questions, findings, and theoretical contribution:

· Strengthened the Introduction and Literature Review to more clearly position political identity as a primed and contested aspect of self that carries both cognitive and relational stakes. We more explicitly connect political identity work to the broader literatures on social identity, self-categorization, and identity verification.

· Clarified the exploratory nature of our research design in the revised Methods and Discussion sections, explicitly framing this as a meaning-making study that applied theory after the fact to help make sense of emergent patterns in the data.

· Added a new summary section (Section 3.5) that explicitly answers our guiding research questions by identifying four distinct conversational strategies—disengagement, negotiation, context adaptation, and information processing—that students use to either avoid or stay engaged in political conversations. This summary helps the reader clearly see how our data respond directly to the research questions.

· Strengthened our theoretical framing in the Discussion (Section 4) by positioning these strategies not as isolated conversational tactics, but as practical expressions of deeper identity work. We explain how these strategies function as recalibration mechanisms in response to primed political identity, drawing on Self-Categorization Theory (SCT) and Identity Control Theory (ICT) as post hoc lenses to interpret these dynamics.

· Expanded Section 5.1 to clarify that these conversational strategies reflect adaptive efforts to manage social, cognitive, and emotional risks—not just to "win" conversations, but to protect relational ties, self-concept, and conversational control when political identity is salient.

· Strengthened Section 5.2 by highlighting the role of identity discomfort as a central barrier to engagement, and by proposing the need for future research on building discomfort tolerance as a foundation for more meaningful political dialogue.

· Added concrete forward-looking implications for research and practice, including the need to test identity discomfort interventions and examine how different social and relational contexts shape avoidance and engagement dynamics.

· Finally, we have decided to change the title to better indicate the nature and findings of the research.

Taken together, these revisions improve the overall coherence and contribution of our manuscript by showing more clearly how the conversational strategies we identified respond to the questions we posed, and why they matter as reflections of deeper identity management processes. We believe the manuscript is now significantly stronger thanks to your push, and we hope you agree that these changes address your concerns.

Issue 2: Claim Social Identity and Emotions are Central—But Don’t Show Direct Evidence

“The authors argue that students may avoid political conservation with ideologically dissimilar others as a form of protecting their social identity. In this argument, emotions and social identity were proposed as key conceptual background. However, no empirical evidence was provided to support this claim."

Our Response

We appreciate this additional critique and fully agree that our earlier framing leaned too heavily on the emotional dynamics of identity without sufficiently demonstrating how consistently these dynamics showed up in the data. Your comment prompted us to take a closer look at how we were balancing our interpretation of emotional and cognitive processes.

In response, we have made two targeted adjustments to better reflect what the data actually support:

· Clarified the Role of Identity Processes: We revised the Results and Discussion to show more clearly how students described monitoring conversational risks, adjusting their behavior, and managing the flow of interactions. While students rarely used explicit identity language themselves, these strategies align well with identity management processes described in Self-Categorization Theory and Identity Control Theory. We now frame these theories more explicitly as post hoc interpretive lenses that help make sense of students’ meaning-making, rather than as predefined explanatory models.

· Calibrated Our Claims About Emotion: We have moderated our claims about emotional drivers, acknowledging that while emotional undercurrents—such as frustration, stress, or discomfort—were present in some participants’ accounts, these themes were not consistently emphasized across the full dataset. Instead, students most often described cognitive strategies such as categorization, risk evaluation, and conversational adjustment. We now describe emotional dynamics as a meaningful but secondary layer, integrated into broader cognitive and social processes, rather than as a primary or universal driver of behavior.

Together, these changes help us present a more balanced interpretation—one that acknowledges the role of emotional dynamics when they arise, but foregrounds the cognitive and social identity management work that students more consistently described. We hope this revised framing more accurately reflects the data and addresses your important concern.

Issue 3: We Mention Emotions Like Stress and Anxiety—But Don’t Prove the Link

“The authors provided emotional explanations for avoiding political conversations as discussing political issues generates negative emotions (stress and anxiety); however, this link has not been tested.”

Our Response

Thank you again for pushing us to sharpen our claims. To avoid redundancy, we refer to our earlier responses where we already addressed how we clarified the role of emotional dynamics in relation to cognitive and identity processes. Here, we add a more specific acknowledgment of the limitations of our data regarding emotional outcomes like stress or anxiety.

While several participants did reference feelings such as frustration or emotional fatigue, these accounts were uneven and often secondary to the more consistent cognitive strategies students described—such as categorizing conversation partners, evaluating risks, or managing interactional flow. As we explained in our earlier responses, we have revised the manuscript to position emotional dynamics as a possible subtext rather than a primary theme.

In particular, we have:

· Clarified in the Methods that emotional codes were part of our early analysis but did not hold up as a standalone theme due to inconsistent salience across the dataset.

· Noted in the Results that while emotions were mentioned by some participants (e.g., feelings of discomfort or fatigue), they were not a universal feature of students’ accounts.

· Framed in the Discussion the role of emotions as a likely but uneven substratum to the more observable cognitive and identity management work students described.

We fully agree with the reviewer that our data do not “test” or prove the emotional link, and we now clearly state that in the manuscript. We frame this instead as a promising direction for future research—suggesting that emotional and identity discomfort may interact, but require more direct investigation.

We appreciate this push to better align our claims with what our data can—and cannot—support. We believe these revisions make for a stronger, more disciplined contribution.

Issue 4: Are the Dynamics Unique to Political Identity

"Is this effect domain specific (political identity) or domain general (social identity in general)? The authors claimed that people tend to disengage as a means of preserving self-consistency and emotional equilibrium. Wouldn’t this be the case for any other identity?"

Our Position

Thank you for raising this important distinction. We appreciate the opportunity to clarify how our revised manuscript now addresses this concern.

While it is certainly plausible that the strategies students described could apply to a variety of identity-relevant contexts beyond politics, we worked in our revisions to strengthen the case for why political identity, in particular, appears to be uniquely situationally volatile, socially risky, and interactionally consequential in the current sociopolitical climate. Specifically, in the Primed Political Identities section of the Discussion, we expanded our analysis to draw more explicitly on Self-Categorization Theory (SCT) to explain how political identity becomes situationally primed through environmental, relational, and conversational cues.

We argue that political identity—unlike many other social identities (e.g., being a student, sibling, or friend)—is more likely to become situationally primed and interactionally salient in moments of social tension or perceived disagreement, especially in the current polarized climate. This volatility stems from the high visibility, divisiveness, and personal stakes often attached to political identity in public and private life today. Once primed, students describe navigating these moments

through identity management strategies aimed at protecting or stabilizing their political self-concept in relation to others.

In the revised manuscript, we also clarify that these processes are not static but highly dynamic, emerging when students anticipate identity-relevant risks or when conversational dynamics signal potential misalignment. We argue that Identity Control Theory (ICT) provides a useful lens to interpret these recalibration efforts, as students work to maintain coherence between their internal political identity standards and the feedback they receive from others in real time.

Finally, while we acknowledge that similar identity management strategies may occur in other domains of social life, we now more explicitly argue that the heightened visibility, divisiveness, and potential social consequences of political identity make it especially prone to these types of verification struggles. Drawing on SCT’s emphasis on contextual salience and social consequence, we make the case that political identity—more than many other everyday identities—tends to flare into salience in ways that require students to manage conversational and relational risk.

We believe these additions strengthen the manuscript’s theoretical precision and help clarify why political identity, specifically, is the focus of our analysis. We hope these revisions address the reviewer’s concern.

Issue 5: Clarifying the Theoretical Relevance of the Four Strategies

"Also, the four strategies people use in political conversation do not seem to provide answers to the main research questions that authors discussed in the manuscript."

"Also, it might be helpful if the authors provide more convincing reasons why the findings are important. For example, how do the four strategies relate to identity control theory?"

Our Response:

We appreciate this important push to strengthen the theoretical integration of our findings. In direct response, we have revised the Discussion to more clearly position the four identified strategies—disengagement, negotiation, context adaptation, and information processing—as concrete expressions of real-time identity management processes.

As we noted earlier in Response 1, these strategies now serve as our clearest answers to the two guiding research questions: (1) why students disengage from political conversations, and (2) under what conditions they remain open to engagement. To avoid redundancy, we simply note here that this re-alignment of findings and framing is reflected in the new Results Summary (Section 3.5) and expanded Discussion (Section 4).

More specifically, we have strengthened the Discussion by linking these strategies more explicitly to Self-Categorization Theory (SCT) and Identity Control Theory (ICT). We outline how SCT helps explain when political identity becomes psychologically and socially primed, triggering students to monitor social cues for conversational risk. We then use ICT to explain the corrective actions that follow, showing how the four strategies reflect dynamic identity regulation efforts aimed at preserving alignment between internal identity standards and external feedback.

We further clarify how these four strategies map onto three broader identity verification mechanisms—avoidance, modification, and reinforcement—demonstrating that they are not isolated conversational moves, but adaptive responses to perceived social and identity risk. We also emphasize that these strategies help explain not only how students manage political disagreement, but why certain conversational settings or relationships make engagement feel more or less possible.

Taken together, these changes strengthen the link between our empirical findings, our research questions, and the theoretical contributions of the study. We hope this revised narrative now offers

a more convincing and coherent explanation of why these strategies matter and how they advance identity-based theories of political communication.

Issue 6: Deepening Engagement with Identity Theories and Prior Literature

"As the authors base their research on social identity theory, related literature needs to be discussed more in depth and results need to be discussed related to prior findings."

Our Response

We thank Reviewer 1 for encouraging us to more fully engage with the identity literature and clarify the theoretical contribution of our findings. This was an extremely helpful push that shaped several key improvements across the manuscript. Specifically, we made the following changes:

· Introduction: We strengthened the introduction to better position our study within broader debates on political identity salience, polarization, and dialogue across divides. While our primary focus remains on identity processes, we now briefly acknowledge competing explanations (e.g., moral or epistemic asymmetries) before situating our study within a social psychological and interactional framework informed by identity theory.

· Literature Review: We expanded our review of both Self-Categorization Theory (SCT) and Identity Control Theory (ICT), providing clearer definitions and showing how they complement each other in explaining when political identities become salient (SCT) and how individuals regulate identity alignment in response (ICT). We also engage more fully with prior empirical work on conversational avoidance, identity threats, and polarization to situate our findings within existing research.

· Discussion: We now more explicitly map the four conversational strategies identified in our data—disengagement, negotiation, context adaptation, and information processing—onto the avoidance, modification, and reinforcement mechanisms described by ICT. We also clarify how SCT helps explain the priming of political identity as situationally volatile and socially consequential. These additions offer a more complete theoretical account of both the triggers and responses involved in managing political identity in conversation.

We believe these revisions make our theoretical contribution much clearer and more robust. We again thank Reviewer 1 for this valuable push, which helped us deepen the conceptual framing and sharpen the contribution of the study.

Issue 7: Why Do Our Findings Matter

"Also, it might be helpful if the authors provide more convincing reasons why the findings are important. For example, how do the four strategies relate to identity control theory?"

Our Response

We thank Reviewer 1 for this critical push to clarify the practical and theoretical significance of our findings. In the revised manuscript, particularly in Sections 5.1 and 5.2, we have strengthened our case for why these conversational strategies matter both empirically and conceptually.

We argue that the four strategies identified in our study—disengagement, negotiation, context adaptation, and information processing—are not just conversational habits or surface-level techniques. Instead, we frame them as socially learned identity management practices that students use to navigate the cognitive and social risks of political disagreement and the threats these pose to their political identities.

By grounding these strategies in Self-Categorization Theory (SCT) and Identity Control Theory (ICT), we show how they operate as dynamic identity regulation mechanisms that help students manage self-presentation, relational tension, and cognitive coherence when political identity

becomes salient or threatened. SCT helps explain how political identity is situationally activated by environmental and conversational cues, while ICT provides a framework for understanding how students work to re-establish alignment between their internal identity standards and the social feedback they receive.

We have strengthened the Discussion section to clarify that these strategies—while effective in reducing immediate discomfort—may also reinforce longer-term patterns of avoidance and ideological segregation if left unexamined. This insight moves beyond skill-based or civility-focused models of political dialogue by drawing attention to the deeper identity discomfort that often drives avoidance in the first place.

Recognizing these patterns creates space for future research and practice aimed at not only improving conversational skills, but also fostering greater tolerance for identity discomfort—a capacity we see as critical for sustaining engagement across ideological divides in today’s polarized social climate.

We again thank Reviewer 1 for prompting us to clarify this contribution, which we believe is now more compellingly communicated in the revised manuscript.

Reviewer 2 Report

Comments and Suggestions for Authors

I congratulate you on the topic and the method chosen, it's really important for many societies at this historical moment in politics.

Although it seems to be an exploratory research, it is interesting and opens the door to other studies and is worth publishing for many reasons.

However, the there are some questions that must be resolved:

  1. What other previous studies have been carried out in this line or that address some of the aspects you studied in the last years 10 years, specially in the last 8 years, after the first time Trump was president in 2017?
  2.  The theoretical framework must be updated.
  3.  The research objectives or questions are not easily found; they should be more clearly defined, identified in one of the first sections of the article.
  4. I miss the conclusions of the results. The rest is interesting (limits, future researchs...), but you have to write about the conclusions of your study after your results (you can start with a summary of the results that will help people who will read your paper) and a discussion with the new recent references you will include. 

I will read with interest the results of implementing these essential improvements. 

Author Response

Issue 1: Recent Literature Need Updating

“What other previous studies have been carried out in this line or that address some of the aspects you studied in the last years 10 years, specially in the last 8 years, after the first time Trump was president in 2017? The theoretical framework must be updated.”

Our Response

We appreciate this recommendation and fully agree that strengthening our engagement with recent scholarship is critical to the relevance and impact of this study. In response, we have substantially expanded the literature review, discussion, and conclusion to more explicitly situate our work in the post-2017 political landscape.

Specifically, we have now included recent studies published in the last eight years that speak to identity management, polarization, and the challenges of political discourse in the Trump and post-Trump era. These include, but are not limited to, recent contributions by Ruckelshaus (2022), Hernández et al. (2021), Perry (2022), Benkler, Faris, and Roberts (2018), Bennett and Livingston (2018), and McIntyre (2018). These additions strengthen our theoretical framing by showing how our findings connect to current debates about affective polarization, epistemic fragmentation, and identity threat in today’s political climate.

At the same time, we have retained earlier foundational scholarship—such as Poole and Rosenthal’s (1984) long-term analysis of U.S. polarization—because we believe that understanding today’s divisions requires both historical and contemporary perspectives. We have made this balance more explicit in the revised manuscript to reflect both the deep roots and recent intensification of political identity dynamics.

Finally, we have expanded the conclusion to summarize the relevance of these recent literatures, explicitly connecting our findings to contemporary concerns about affective polarization, identity discomfort, and relational risks in political dialogue. This not only clarifies the stakes of our findings but also situates them within ongoing interdisciplinary conversations about how political identity shapes discourse in the post-2017 era.

We hope these additions and clarifications fully address Reviewer 2’s valuable push to update and contextualize our theoretical framework.

Issue 2: Theoretical Framework

“The theoretical framework must be updated.”

Our Position

Thank you for highlighting this important issue. We fully agree that the theoretical framework in the earlier version was underdeveloped and needed to be strengthened. In revising the manuscript, we recognized that our constructivist and interpretivist orientation had not come through as clearly as intended. We have now made this orientation explicit in the revised Introduction, Methods, and Discussion, emphasizing that our aim was not to test predefined hypotheses, but to build understanding from student meaning-making and then apply theory to help explain the patterns we observed.

We have also substantially expanded and clarified the theoretical framing of the study. Specifically, we now provide a more thorough discussion of identity theory, drawing on both Self-Categorization Theory (SCT) and Identity Control Theory (ICT) to explain how students navigate political conversations when political identity becomes situationally primed. We argue that SCT helps explain when and why political identity becomes socially and psychologically salient, while ICT provides a lens for understanding how students work to restore identity stability when they experience potential or actual identity misalignment.

In addition, we have strengthened the literature review to better connect our study to broader research on political identity, polarization, and identity management. We now more clearly position our work in relation to both classic and contemporary research, including recent studies published in the last eight years that address the intensification of identity-based polarization in the Trump and post-Trump era. These additions make our theoretical grounding more relevant to the current political context while maintaining connection to foundational identity theory.

We believe these revisions significantly improve the clarity, coherence, and relevance of the manuscript’s theoretical foundation. Thank you again for prompting these improvements.

Issue 3: Research objectives or questions not clearly stated enough

“The research objectives or questions are not easily found; they should be more clearly defined, identified in one of the first sections of the article.”

Our Response:

We appreciate this helpful observation and have taken steps to address it directly. In the revised manuscript, we now clearly state our two guiding research questions in the introduction:

1. How do college students describe their experiences of avoiding or engaging in political conversations across ideological divides?

2. Under what conditions do they feel more open to having these conversations?

In addition, we have worked to clarify the overall framing and epistemological orientation of the study. Specifically, we now more explicitly communicate that this is an exploratory, constructivist project grounded in students’ own meaning-making about their conversational experiences. Our focus is on how students make sense of these interactions, not on establishing causal explanations or objective claims about what “works” in political dialogue.

We have also revised the opening of the Discussion to connect these research questions more directly to our findings, showing how the four identified conversational strategies—disengagement, negotiation, context adaptation, and information processing—reflect the practical ways students navigate these challenges. Importantly, we argue that these strategies are best understood not as surface-level conversation techniques, but as deeper forms of identity work—

socially patterned responses to the cognitive and relational risks students experience when navigating political disagreement. By positioning these strategies as identity management processes, we clarify their theoretical significance and contribute to broader conversations about identity regulation in politically charged social interactions. We hope these improvements resolve the reviewer’s concern by making the purpose, framing, and contribution of the study clearer and more accessible to readers.

Issue 4: Conclusions Are Missing or Underdeveloped

“I miss the conclusions of the results. The rest is interesting (limits, future researchs...), but you have to write about the conclusions of your study after your results (you can start with a summary of the results that will help people who will read your paper) and a discussion with the new recent references you will include.”

Our Response

Thank you for this excellent suggestion. We fully agree that the original submission lacked a clear, accessible summary of what our findings revealed before moving into theoretical interpretation. Your comment prompted us to reflect more intentionally on how we communicate the overall contribution of the study.

In response, we have added a new bridging section at the end of the Results titled “3.5. Summary: A Practical Inventory of Student Strategies.” This section synthesizes the four strategies—disengagement, negotiation, context adaptation, and information processing—and clarifies what they reveal about how students navigate the social, cognitive, and relational stakes of political disagreement. This helps orient readers to the “big picture” of the findings before transitioning into deeper theoretical analysis in the Discussion.

In addition, we have expanded the concluding sections of the paper to offer not only a summary but also a clear interpretation of why these strategies matter. In Section 5.1, we frame these strategies as expressions of deeper identity management processes, showing how students respond to identity-related risks when navigating politically charged conversations.

We also strengthen the theoretical relevance of our conclusions by engaging with recent scholarship (e.g., Mezirow, 2000; Zúñiga et al., 2007; Haidt, 2012; Bruneau & Cikara, 2017; Levendusky, 2023), positioning our findings within current debates on dialogue, polarization, and identity. Finally, we outline practical implications for research, education, and practice—emphasizing, in particular, the need to build students’ capacity for managing identity discomfort in politically divided settings.

We believe these revisions result in a stronger and more meaningful conclusion that both synthesizes the findings and highlights their theoretical and practical significance. We appreciate the reviewer’s helpful push, which allowed us to clarify the broader value of our work.

Reviewer 3 Report

Comments and Suggestions for Authors

Overall, I have no problem with the study as conducted and reported. The methodology is sound, and the research supports the conclusions. My concern is the larger social context. The study treats both sides of polarized views as equally legitimate. This would be accurate if both sides agreed on basic measures of reality; they do not. Today's conservatism in the United States is not genuinely conservative, but pseudo-conservative (fascist). In the history of the US, liberal and conservative sides at least agreed on the facts of, for example, the economy, immigration, civil rights, and so on. They differed on what policies should follow from those facts. Today, the pseudo-conservatives (fascists) reject any fact they don't like as "fake news," following the mantra of their leader, Donald Trump, various online influencers, and conspiracy fantasists. They reject science. They reject logic and reason. They accept only what provides emotional gratification. In contrast, mainstream and liberal views premise their perspectives on science, logic, and reason. Discussion has become impossible because the pseudo-conservatives, quite simply, reject reality. With completely different standards of reality, no discussion is possible. I think this study needs to account for this, that one side starts with the real world, and the other rejects it. The mainstream and liberal side can be persuaded with better facts and logic. The fascists only by pronouncements from the leaders they admire.

Author Response

Issue 1: Concerns About Ideological Asymmetries and Broader Political Context

"My concern is the larger social context. The study treats both sides of polarized views as equally legitimate."

"Today's conservatism in the United States is not genuinely conservative, but pseudo-conservative (fascist)."

"Discussion has become impossible because the pseudo-conservatives, quite simply, reject reality. With completely different standards of reality, no discussion is possible."

"I think this study needs to account for this, that one side starts with the real world, and the other rejects it."

Our Response

We appreciate Reviewer 3’s thoughtful engagement with the broader societal stakes of this work. We recognize and agree that the current political climate in the United States presents asymmetrical challenges for democratic dialogue, particularly in relation to rising concerns about misinformation, conspiratorial thinking, and the erosion of shared standards of truth in what some have called a “post-truth” era (Benkler, Faris, & Roberts, 2018; Bennett & Livingston, 2018; McIntyre, 2018).

We understand the reviewer’s concern that our study might appear to treat all ideological positions as equally grounded in fact or reason. While this is not our intention, we acknowledge that our social constructionist, participant-centered approach focuses on how students themselves describe navigating these divides—regardless of whether their conversation partners, or even the students themselves, operate from factually consistent frameworks. Our analysis centers on how students perceive and manage conversational risks, rather than adjudicating which ideological positions are more legitimate in an objective sense.

That said, we recognize that students’ struggles to sustain dialogue are likely shaped by these larger asymmetries in how different groups relate to facts, authority, and epistemic standards. We have therefore added language in our introduction acknowledging these broader structural and epistemic challenges. While our data do not allow us to resolve these societal-level issues, we hope this addition clarifies how our micro-level focus on student experience sits within this wider and deeply consequential political context.

We hope this addition strikes an appropriate balance—honoring the reviewer’s concern without overstepping the boundaries of our data or re-framing the study as taking a normative position on partisan legitimacy.

Round 2

Reviewer 1 Report

Comments and Suggestions for Authors

I appreciate the effort the authors put into revising manuscript. I also noticed that the authors updated the front end with some additional literature review on theories explaining political identity. However, there seems to be still remaining questions which I describe below:

  • I understand the current article is not taking a hypothetico-deductive approach. However, what appears to miss in this article is “so what?”
  • It might be helpful to address how the findings contribute to future research. For example, proposing future research agenda based on the findings.
  • It is helpful to discuss theoretical lenses through which political identity can be studied: social identity, social categorization, and control theories. What is missing is how the authors build their own theorization based on those. How are they related or not related to their own proposition? What is additional knowledge the current study contributes to?
  • The authors claimed that the current study explores two specific questions: (1) How do college students describe their experiences of avoiding or engaging in political conversations across ideological divides? and (2) Under what conditions do they feel more open to engaging in these conversations? I could have missed it but I do not see if these two questions were tested.

Author Response

Comment 1: The authors claimed that the current study explores two specific questions: (1) How do college students describe their experiences of avoiding or engaging in political conversations across ideological divides? and (2) Under what conditions do they feel more open to engaging in these conversations? I could have missed it but I do not see if these two questions were tested.

Our Response: We appreciate the comment. While we do believe that the existing manuscript had already addressed these questions in both structure and substance, we recognize that this may not have been sufficiently foregrounded for all readers. To enhance clarity, we have added explicit statements in three strategic locations:

  • In the introduction to the results section (Section 3), we clarify how the identification and interpretation of the four themes were guided by the research questions, which served as the analytic lens through which we examined students’ meaning-making practices.
  • At the end of the Data Analysis subsection (Section 2.4), we further emphasize that the analysis was interpretively structured by the research questions, with the goal of illuminating how students make sense of political engagement rather than testing fixed propositions.
  • In the opening paragraph of the Conclusion (Section 5), we reiterate the study’s two guiding research questions and explicitly frame the discussion of our findings as a direct response to them.

We hope that these clarifications will ensure the connections between our findings and the research questions are now unmistakable to readers approaching the paper from a variety of perspectives. Note: See changes in Blue.

Comment 2: It is helpful to discuss theoretical lenses through which political identity can be studied: social identity, social categorization, and control theories. What is missing is how the authors build their own theorization based on those. How are they related or not related to their own proposition? What is additional knowledge the current study contributes to?

Our Response: We thank the reviewer for the suggestion to clarify how our study builds its own theorization based on the conceptual lenses employed. But we would like to gently push back.

Our analytic approach is grounded in a constructionist epistemology. From this standpoint, theory is not treated as a fixed scaffolding imposed prior to analysis, but as a set of interpretive tools brought into reflexive dialogue with participants’ own meaning-making (Charmaz, 2014; Hacking, 1999). While some research traditions foreground theory-building in the literature review, doing so in the context of constructionist inquiry risks prematurely narrowing the analytic gaze and reifying categories that should remain open to contestation.

We included the discussion of Social Identity Theory (SIT), Identity Control Theory (ICT), and Self-Categorization Theory (SCT) in the literature review as a concession to both our quantitatively trained coauthor and your earlier feedback. However, from this methodological perspective, this addition is redundant at best and epistemologically inconsistent at worst. Theory in constructionist work is not a prerequisite grid but a generative resource applied in the course of interpretation. In this spirit, we use SCT and ICT in the Discussion section to make sense of how students navigate political disagreement as a form of identity work. Specifically, SCT helps account for the situational activation of political identity, particularly in socially complex or ideologically ambiguous spaces. And ICT illuminates the self-regulatory processes participants described as they attempted to maintain identity coherence amid interpersonal friction or social threat.

These frameworks are not simply referenced—they are actively used to interpret participants’ narratives and to theorize political discourse as a form of everyday identity negotiation. We believe this interpretive synthesis, developed through analysis rather than front-loaded in the literature review, constitutes a meaningful theoretical contribution.

References:
Charmaz, K. (2014). Constructing Grounded Theory (2nd ed.). Sage.
Hacking, I. (1999). The Social Construction of What? Harvard University Press.

Comment 3: I understand the current article is not taking a hypothetico-deductive approach. However, what appears to miss in this article is “so what?” It might be helpful to address how the findings contribute to future research. For example, proposing future research agenda based on the findings.

Our Response: To address this, we have substantially expanded the Limitations and Directions for Future Research section. In addition to acknowledging the need for broader and more diverse samples, we now identify several specific research directions that emerge from our findings. First, we suggest that the four conversational strategies identified in this study—Disengagement, Negotiation, Context Adaptation, and Information Processing—can be understood not as discrete techniques but as surface-level expressions of deeper identity regulation. Building on our analytic framework, we propose that three underlying identity processes—identity salience, coherence, and feedback management—may offer a useful lens for future inquiry. These could be examined across different ideological intensities, relational settings (e.g., classrooms, families, online platforms), and sociocultural contexts.

Second, we call for research into identity discomfort tolerance: that is, the capacity to remain present in situations where one’s identity feels challenged, misrecognized, or unsettled. Our findings suggest that many students engaged in avoidance-oriented strategies that protected short-term emotional coherence but potentially foreclosed opportunities for learning and relationship-building. Future work might explore whether alternative forms of identity work—those that can hold ambiguity, friction, or temporary incoherence—could support more relationally grounded political engagement. We also highlight the potential for pedagogical and civic interventions that cultivate dialogic resilience and identity flexibility. Finally, we emphasize the importance of affective dynamics. While not always foregrounded, emotional responses such as anxiety, shame, and fear appeared to play a critical role in shaping conversational choices. We now explicitly invite future research to examine how individuals manage these emotional dynamics in real time—through suppression, redirection, or relational repair—and how such regulation interacts with broader identity processes.

Together, we hope these additions clarify both the “so what” of our findings and the broader research agenda they open up—not only for scholars of political discourse and identity, but also for educators, facilitators, and practitioners working to support more constructive engagement across divides. Note: See changes in blue.

Round 3

Reviewer 1 Report

Comments and Suggestions for Authors

The revised manuscript answered most of my previous concerns.